# Single cell regulatory landscape of the mouse kidney highlights cellular differentiation programs and disease targets

Zhen Miao [1,2,3,8], Michael S. Balzer [1,2,8], Ziyuan Ma [1,2,8], Hongbo Liu[1,2], Junnan Wu [1,2], Rojesh Shrestha [1,2], Tamas Aranyi[1,2], Amy Kwan[4], Ayano Kondo [4], Marco Pontoglio [5], Junhyong Kim[6], Mingyao Li [7], Klaus H. Kaestner[2,4] & Katalin Susztak [1,2,4✉]

Determining the epigenetic program that generates unique cell types in the kidney is critical for understanding cell-type heterogeneity during tissue homeostasis and injury response. Here, we profile open chromatin and gene expression in developing and adult mouse kidneys at single cell resolution. We show critical reliance of gene expression on distal regulatory elements (enhancers). We reveal key cell type-specific transcription factors and major gene-regulatory circuits for kidney cells. Dynamic chromatin and expression changes during nephron progenitor differentiation demonstrates that podocyte commitment occurs early and is associated with sustained *Foxl1* expression. Renal tubule cells follow a more complex differentiation, where *Hfn4a* is associated with proximal and *Tfap2b* with distal fate. Mapping single nucleotide variants associated with human kidney disease implicates critical cell types, developmental stages, genes, and regulatory mechanisms. The single cell multi-omics atlas reveals key chromatin remodeling events and gene expression dynamics associated with kidney development.

[1] Renal, Electrolyte, and Hypertension Division, Department of Medicine, University of Pennsylvania, Perelman School of Medicine, Philadelphia, PA, USA. [2] Institute for Diabetes, Obesity, and Metabolism, University of Pennsylvania, Perelman School of Medicine, Philadelphia, PA, USA. [3] Graduate Group in Genomics and Computational Biology, University of Pennsylvania, Perelman School of Medicine, Philadelphia, PA, USA. [4] Department of Genetics, University of Pennsylvania, Perelman School of Medicine, Philadelphia, PA, USA. [5] Epigenetics and Development Laboratory, Université de Paris Inserm U1151/CNRS UMR 8253, Institut Necker Enfants Malades, Paris, France. [6] Department of Biology, University of Pennsylvania, Philadelphia, PA, USA. [7] Department of Epidemiology and Biostatistics, University of Pennsylvania, Perelman School of Medicine, Philadelphia, PA, USA. [8] These authors contributed equally: Zhen Miao, Michael S. Balzer, Ziyuan Ma. ✉email: ksusztak@pennmedicine.upenn.edu

The mammalian kidney maintains fluid, electrolyte, and metabolite balance of the body and plays an essential role in blood pressure regulation and red blood cell homeostasis. The human kidney makes roughly 180 liters of primary filtrate each day that is then reabsorbed and modified by a long tubule segment. To perform this highly choreographed and sophisticated function, the kidney contains close to 20 highly specialized epithelial cells. The renal glomerulus acts as a 60 kD size-selective filter. The proximal part of the tubules is responsible for reclaiming more than 70% of the primary filtrate, which is done via unregulated active and passive paracellular transport[1], while the loop of Henle plays an important role in concentrating the urine. The distal convoluted tubule is critical for regulated sodium reabsorption. The last segment of kidney tubules is the collecting duct, where the final concentration of the urine is determined via regulation of water channels, acid, or base secretion. Understanding the development of these diverse cell types in the kidney is essential to understand kidney homeostasis, disease, and regeneration.

The mammalian kidney develops from the intermediate mesoderm via a complex interaction between the ureteric bud and the metanephric mesenchyme[2]. In the mouse kidney, Six2 marks the self-renewing nephron progenitor population[3]. The nephron progenitors commit and undergo a mesenchymal-to-epithelial transformation giving rise to the renal vesicle[3]. The renal vesicle then undergoes segmentation and elongation, giving rise to epithelia from the podocytes to the distal convoluted tubules, while the ureteric bud becomes the collecting duct. Unbiased and hypothesis-driven studies have highlighted critical stages and drivers of early kidney development[4], that have been essential for the development of an in vitro kidney organoid differentiation protocols[5–7]. However, cells in organoids are still poorly differentiated, improving cellular differentiation and maturation of these structures remains a major challenge[8]. Thus, the understanding of late kidney development, especially the cell type-specific driver transcription factors (TFs) is of great importance[9–11]. Alteration in Wnt, Notch, Bmp, and Egf signaling significantly impacts cellular differentiation, but only a handful of TFs that directly drive the differentiation of distinct segments have been identified, such as *Pou3f3*, *Lhx1*, *Irx2*, *Foxc2*, and *Mafb*[12]. Further understanding of the terminal differentiation program could aid the understanding of kidney disease development.

While single cell RNA sequencing (scRNA-seq) has improved our understanding of kidney development in mice and humans[9,10,13,14], it provides limited information of TFs, which are usually expressed at low levels. Equally difficult is to understand how genes are regulated from scRNA-seq data alone. Chromatin state profiles, on the other hand, determine the gene expression potential and can pinpoint the availability of TF binding sites. Together with gene expression, open chromatin profiles can define the gene regulatory logic, which is the fundamental element of cell identity. However, there is a scarcity of open chromatin information by Assay for Transposase-Accessible Chromatin using sequencing (ATAC-seq) or chromatin immunoprecipitation (ChIP) data by ChIP-seq related to kidney postnatal development. In addition, epigenetic changes observed in bulk analyses mostly represent changes in cell composition, rather than cell type-specific changes[15], making it challenging to interpret bulk ATAC-seq data. The publicly available mouse kidney ATAC-seq data contain limited number of cells from adult tissues, and therefore do not provide insight into chromatin changes during development[16,17].

Here, we generate a single cell open chromatin and corresponding expression survey for the developing and adult mouse kidney, which is available to the community via searchable websites (susztaklab.com/developing_adult_kidney/snATAC/ for snATAC-seq data, susztaklab.com/developing_adult_kidney/scRNA/ for scRNA-seq data, and susztaklab.com/developing_adult_kidney/igv/ for IGV view of peak tracks). Using this atlas, we produce an epigenome-based classification of developing and mature cells and defined cell type-specific regulatory networks. We also investigate key TFs and cell–cell interactions associated with developmental cellular transitions. Finally, we use the single cell open chromatin information to pinpoint putative target genes and cell types of several chronic kidney disease noncoding genome-wide association study (GWAS) loci.

## Results

**Single cell accessible chromatin landscape of the developing and adult mouse kidneys.** To characterize the accessible chromatin landscape of the developing and adult mouse kidneys at single cell resolution, we performed single nuclei ATAC-seq (snATAC-seq) on kidneys of mice on postnatal day 0 (P0), at 3 and 8 weeks (P21 and P56) of age (Fig. 1a). In mice at birth, the nephron progenitors are still present and nephrons are formed until around day 21[13,18]. In parallel, we also performed bulk (whole kidney) ATAC-seq analysis at matched developmental stages. Following sequencing, we aggregated all high-quality mapped reads in each sample irrespective of barcode. The combined snATAC-seq dataset from all samples showed the expected insert size periodicity (Supplementary Fig. 1a) with a strong enrichment of signal at Transcription Start Sites (TSS), indicating high data quality (Supplementary Fig. 1b). The snATAC-seq data showed high concordance with the bulk ATAC data (Spearman correlation coefficient >0.84 between matched stages, see "Methods" section, Supplementary Fig. 1c).

We next revealed cell type annotations from the open chromatin information. After conducting stringent filtering of the number of unique fragments, promoter ratio and mitochondria ratio (see "Methods" section, Supplementary Fig. 2a), we kept 28,316 cells across the samples (Fig. 1b). Cells were then clustered using SnapATAC[19], which binned the whole genome into 5 kb regions to address the sparsity of the data (see "Methods" section). Prior to clustering, we used Harmony[20], an iterative batch correction method, to correct for variability across samples (Supplementary Fig. 2b). Using batch-corrected low dimensional embeddings, we retained 13 clusters, all of which had consistent representation across the number of peaks, samples and read depth profiles (Figs. 1b and S2c, d).

To determine the cell types represented by each cluster, we examined chromatin accessibility around the TSS and gene body regions of the known cell type-specific marker genes[21]. Based on the accessibility of the known marker genes, we identified clusters representing nephron progenitors, endothelial cells, podocytes, proximal tubule segment 1 and segment 3 cells, loop of Henle, distal convoluted tubule, connecting tubule, collecting duct principal cells, collecting duct intercalated cells, stromal, and immune cells. Fig. 1d and Supplmentary Fig. 3) show chromatin accessibility information for key cell type marker genes, such as *Uncx* and *Cited1* for nephron progenitors, *Nphs1* and *Nphs2* for podocytes, *Akr1c21* for both segments of proximal tubules, *Slc34a1* and *Slc5a2* for proximal convoluted tubules (PCT), *Kap* for proximal straight tubules (PST), *Slc12a1* and *Umod* for loop of Henle, *Slc12a3*, and *Pvalb* for distal convoluted tubule, *Trpv5* for connecting tubule, *Aqp2* and *Fxyd4* for principal cells, *Atp6v1g3* and *Atp6v0d2* for intercalated cells, *Egfl7* for endothelial cells, *C1qb* for immune cells and *Col3a1* for different types of stromal cells, respectively[21]. As expected, some clusters such as nephron progenitors and stromal cells were enriched in the developing kidney (P0).

In order to identify cell type-specific open chromatin regions, we conducted peak calling using MACS2[22] on each cell type

separately. The peaks were then merged to obtain a comprehensive open chromatin set. We found that the single nuclei open chromatin set showed good concordance with bulk ATAC-seq samples, with most of the peaks in bulk ATAC-seq data captured by the single nuclei data. On the other hand, single nuclei chromatin accessibility data showed roughly 50% more accessible chromatin peaks (total of 300,693 peaks) than the bulk ATAC-seq data (Fig. 1e, see "Methods" section). As expected, in general, the overlap with bulk samples was greater for common cell types, such as PT and DCT, than rare cells, such as immune cells, indicating that the snATAC-seq data was particularly powerful in identifying open chromatin areas that are accessible in single cell types.

In parallel, we also generated a single cell RNA sequencing (scRNA-seq) atlas for mouse kidney samples at P0 and P56. Rigorous quality control yielded a set of 43,636 single cells (Fig. 1b and Supplementary Data 1). Quality control metrics such as gene counts, UMI counts, and mitochondrial gene percentage along with batch correction results are shown in Supplementary Fig. 4a–d. We obtained 17 clusters by unbiased clustering[23] and batch effect correction (Supplementary Fig. 4e, f). On the basis of marker gene expression, we identified kidney epithelial, immune, and endothelial cells (Fig. 1f and Supplementary Fig. 5a–c), closely resembling the clustering obtained from snATAC-seq analysis. We then conducted differential expression (DE) analysis on the clusters and identified key marker genes for each cell type (Supplementary Data 2). Correcting gene expression matrices for ambient RNA with SoupX[24] yielded very similar clusters. Average gene expression was highly correlated before and after ambient RNA correction (Supplementary Figs. 6 and 7, see "Methods" section). DE genes for cell types retrieved after correction for ambient RNA are accessible in Supplementary Data 3.

To compare the consistency between cluster assignment in the snATAC-seq data and the scRNA-seq data, we next pooled each snATAC cluster and derived gene activity scores for the top 3000 highly variable genes and computed the Pearson's correlation coefficient between each snATAC cluster and scRNA cluster (see "Methods" section). This analysis indicated good concordance between the two datasets (correlation for P0 samples see Fig. 1g, for adult samples see Supplementary Fig. 8). While the correlation between gene expression and inferred gene activity score was high, we noted some differences in cell proportions, which was likely related to the sample preparation-induced cell drop-out (Supplementary Figs. 2d and 5a). Consistent with previous observations that single cell preparations are more biased towards immune cells than single nuclear preparations[25], we noted that the immune cell repertoire was limited in the snATAC-seq dataset; on the other hand, stromal cells were better captured by the nuclear preparation. Further sub-analysis of the stromal cluster indicated a heterogenous stromal cell population in both P0 and adult samples (Supplementary Figs. 9 and 10). Our data are in keeping with recent findings highlighting considerable heterogeneity in the developing kidney interstitium[26]. DE genes for stroma subclusters are shown in Supplementary Data 4. Additional experimental validation is needed to confirm and further delineate stromal cells.

To allow the interactive use of this dataset by the community, we not only made the raw data available but also the processed dataset via our searchable website (susztaklab.com/developing_adult_kidney/snATAC/ for snATAC-seq data, susztaklab.com/developing_adult_kidney/scRNA/ for scRNA-seq data, and susztaklab.com/developing_adult_kidney/igv/ for IGV view of peak tracks). Supplementary Fig. 11 shows an example of the Ace2 locus in the website.

**Characterization of the cell type-specific regulatory landscape.** To characterize different genomic elements captured by

snATAC-seq data, we first stratified the genome into promoters, exons, 5′ and 3′ untranslated regions, introns, and distal regions using the GENCODE annotation[27] (see "Methods" section). We noticed that concordant with bulk ATAC-seq data, most peaks in snATAC-seq data were in regions characterized as distal elements or introns, and relatively small portions (<10%) were in promoter or 5′ untranslated regions (Supplementary Fig. 12a). Moreover, there were more developmental stage-specific distal and intronic regions (Supplementary Fig. 12b), which is consitent with developmental stage-specific DNA demethylation of the kidney[28]. In addition, around half of the open chromatin peaks overlapped with previously published P0 or adult H3K27Ac ChIP-seq signals[29] (Supplementary Fig. 12c, see "Methods" section). Taken together, these results indicated a critical role of enhancer elements.

To study the open chromatin heterogeneity across cell types and developmental stages, we derived a cell type-specific accessible chromatin landscape by conducting pairwise Fisher's exact test for each peak between every cluster (Benjamini–Hochberg adjusted q value ≤ 0.05, see "Methods" section). In total, we identified 60,684 differentially accessible open chromatin peaks (DAPs) across the 13 cell types (Supplementary Data 5 and Fig. 2a). Among these peaks, most showed high specificity for a single cluster. However, we noticed overlaps between the PCT and PST segments-specific peaks, as well as between the loop of Henle and distal convoluted tubule segments, which is consistent with their biological similarities. In addition to the cell type-specific peaks, we also found some cell type-independent open chromatin areas (present across nephron progenitors, podocytes, proximal tubule and loop of Henle cells), likely consisting of basal housekeeping genes and regulatory elements. (Supplementary Fig. 12d).

We noticed that many genes had strong cell type-specific DAPs at their TSS. Other genes, however, had accessible chromatin at their TSS in multiple cell types. For example, Umod, a marker gene for loop of Henle, showed accessible chromatin at TSS in multiple tubule cell types (Supplementary Fig. 12e). Rather than with its TSS, cell type-specific chromatin accessibility of Umod strongly correlated with an upstream peak, which is likely an enhancer region, as indicated by the H3K27Ac ChIP-seq signal. In addition, we noticed enrichment of intronic regions and distal elements (Supplementary Fig. 13a, b) in cell type-specific DAPs, indicating their role in cell type-specific gene regulation.

These observations motivated us to study cis-regulatory elements using the snATAC-seq data and scRNA-seq data. We reasoned that a subset of the cell type-specific regulatory elements should correlate with cell type-specific gene expression. Inspired by Zhu et al.[30], we aligned DAPs and DEGs from our snATAC-seq and scRNA-seq datasets, and inferred the putative regulatory peak-gene pairs by their proximity (see "Methods" section). Such cis-regulatory elements predictions were confirmed by comparing with cis-regulatory elements inferred previously[31], as we recapitulated roughly 20% of elements from their analysis. In addition, our analysis was able to identify several known enhancers such as for Six2 and Slc6a18[31,32] (Supplementary Fig. 13c).

To quantify the contribution of cis-regulatory elements, we analyzed peak co-accessibility patterns using Cicero[33]. Using a heuristic co-accessible score 0.4 as a cutoff, we identified 232,380 and 206,701 cis-regulatory element links in the P0 and adult data, respectively. However, it is worth-noting that the interaction between genomic elements is complex and is not limited to co-accessible peaks.

Given the complex interaction between genomic regions, we next looked into identifying key TFs that occupy the cell type-specific open chromatin regions. Until now, information on cell type-specific TFs in the kidney has been scarce. Therefore, we performed motif enrichment analysis on the cell type-specific

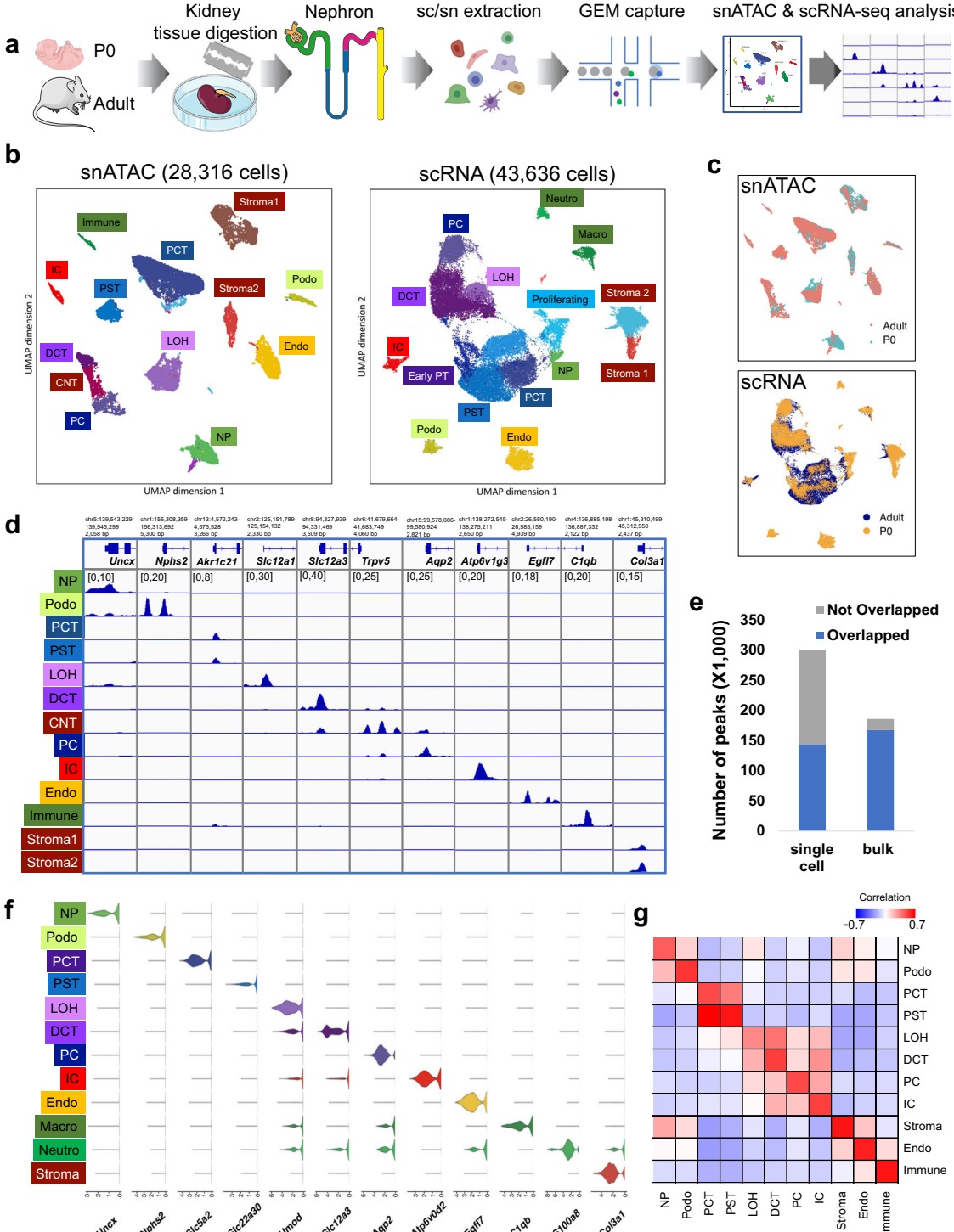

**Fig. 1 snATAC-seq and scRNA-seq identified major cell types in developing and adult mouse kidney. a** Schematics of the study design. Kidneys from P0 and adult mice were processed for snATAC-seq and scRNA-seq followed by data processing and analysis including cell type identification and peak calling; artwork own production and from https://smart.servier.com, license https://creativecommons.org/licenses/by-sa/3.0/). **b** UMAP embeddings of snATAC-seq data and scRNA-seq data. Using marker genes, cells were annotated into nephron progenitors (NP), collecting duct intercalated cells (IC), collecting duct principal cells (PC), proximal convoluted and straight tubule (PCT and PST), loop of Henle (LOH), distal convoluted tubules (DCT), stromal cells (Stroma), podocytes (Podo), endothelial cells (Endo), and immune cells (Immune). In scRNA-seq data, the same cell types were identified, with an additional proliferative population and immune cells were clustered into neutrophils and macrophages. **c** UMAP embeddings of snATAC-seq and scRNA-seq data colored by P0 and adult batches. **d** Genome browser view of read density in each snATAC-seq cluster at cell type marker gene transcription start sites. Additional marker gene examples are shown in Supplementary Fig. 3a. **e** Comparison of peaks identified from snATAC-seq data and bulk ATAC-seq data. Peaks that are identified in both datasets are colored blue, and peaks that are dataset-specific are gray. **f** Violin plots showing cell type-specific gene expression in scRNA-seq data. **g** Heatmap showing Pearson's correlation coefficients between snATAC-seq gene activity scores and gene expression values in P0 data. Each row represents a cell type in scRNA-seq data and each column represents a cell type in snATAC-seq data. The correlation of the adult dataset is shown in Supplementary Fig. 3b.

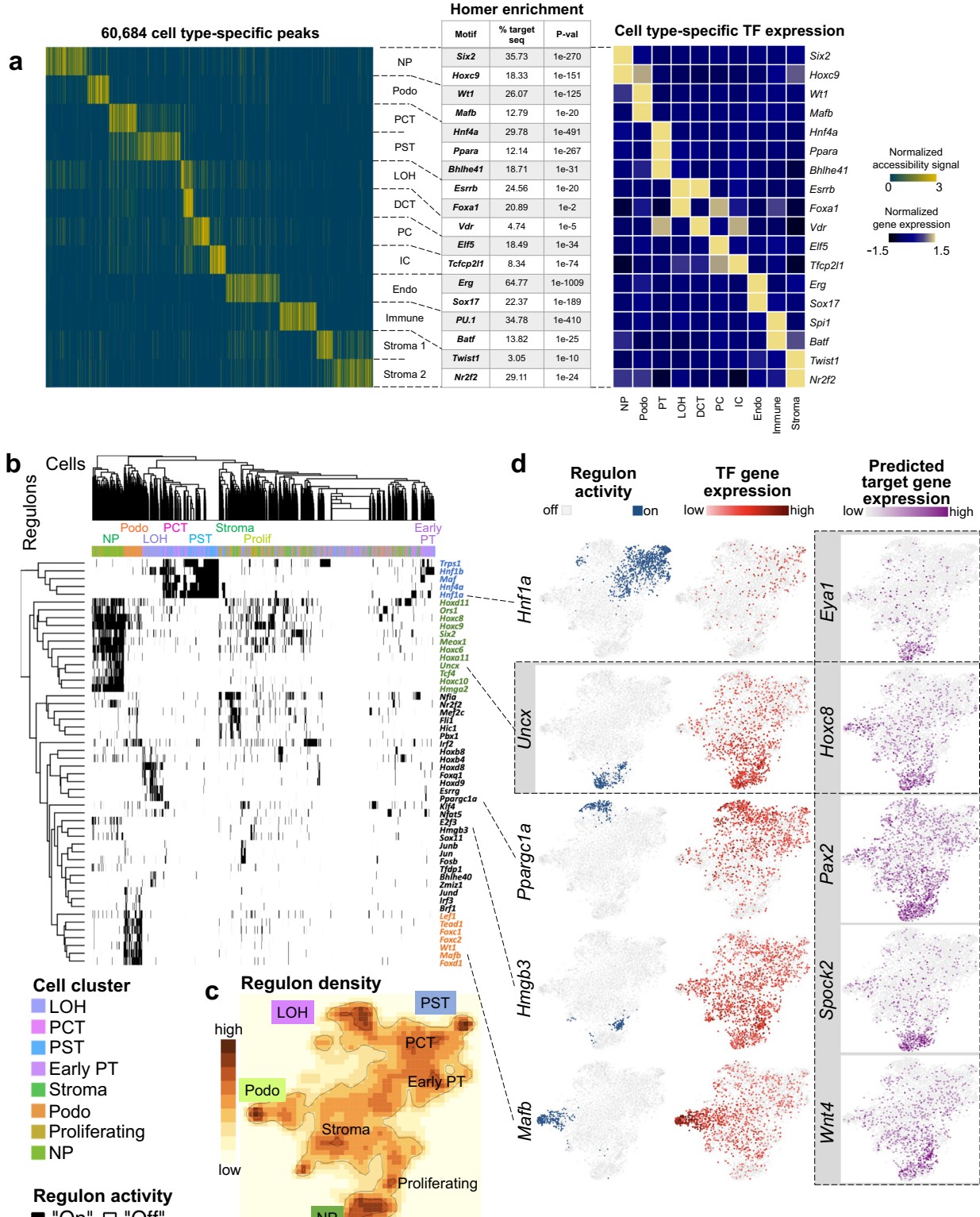

**Fig. 2 Cell type-specific gene regulatory landscape of the mouse kidney. a** Left panel: Heatmap showing all cell type-specific differentially accessible peaks (DAPs) (yellow: open chromatin, blue: closed chromatin) (peak loci are provided in Supplementary Data 5). Middle panel: Examples of cell type-specific motif enrichment analysis using Homer (full results are shown in Supplementary Data 6). Right panel: TF expression *z*-score heatmap that corresponds to the motif enrichment in each cell type. **b** Heatmap of cell type-specific regulons, as inferred by SCENIC algorithm. Regulon activity was binarized to "on" (black) or "off" (white). **c** tSNE representation of regulon density as a surrogate for stability of regulon states, as inferred by SCENIC algorithm. **d** tSNE depiction of regulon activity ("on-blue", "off-gray") and TF gene expression (red scale) of exemplary regulons for proximal tubule (*Hnf1a*), nephron progenitors (*Uncx*), loop of Henle (*Ppargc1a*), proliferating cells (*Hmgb3*), and podocytes (*Mafb*). Examples of target gene expression of the *Uncx* regulon (*Eye1*, *Hoxc8*, *Pax2*, *Spock2*, and *Wnt4*) are shown in purple scale. Expression of target genes of *Hnf1a*, *Six2*, *Ppargc1a*, *Hmgb3*, and *Mafb* is shown in Supplementary Fig. 15a.

open chromatin regions using HOMER (see "Methods" section)[34]. The full list of cell type-specific TF binding motifs is shown in Supplementary Data 6. Since several TFs have identical or similar binding sequences, we next correlated motif enrichment with scRNA-seq TF expression. Using this combined motif enrichment and gene expression approach, we have defined the mouse kidney cell type-specific TF landscape. Examples include *Six2* and *Hoxc9* in nephron progenitors, *Wt1* and *Mafb* in podocytes, *Hnf4a*, *Ppara*, and *Bhle41* in proximal tubules, *Esrrb* and *Foxa1* in loop of Henle, *Vdr* in distal convoluted tubule, *Elf5* in principal cells, *Tcfcp2l1* in intercalated cells, *Erg* and *Sox17* in endothelial cells, *Spi1* and *Batf* in immune cells, and *Twist1* and *Nr2f2* in stromal cells (Fig. 2a and Supplementary Fig. 14).

In order to study the putative target genes of TFs, we examined TF regulon activity using Single-Cell rEgulatory Network Inference and Clustering (SCENIC)[35]. SCENIC was designed to reveal TF-centered gene co-expression networks. By inferring a gene correlation network followed by motif-based filtration, SCENIC keeps only potential direct targets of each TF as modules (regulons). The activity of each regulon in each cell was quantified and then binarized to "on" or "off" based on activity distribution across cells (see "Methods" section). SCENIC was also able to conduct clustering based on the regulon states of each cell. SCENIC results (Fig. 2b–d) indicated strong enrichment in *Trps1*, *Hnf1b*, *Maf*, *Hnf1a*, and *Hnf4a* regulon activity in proximal tubules, *Hmga2*, *Hoxc6*, *Hoxd11*, *Meox1*, *Six2*, *Tcf4*, and *Uncx* in nephron progenitors, *Esrrg*, and *Ppargc1a* in loop of Henle, *Hmgb3* in proliferating cells and *Foxc1*, *Foxc2*, *Foxd1*, *Lef1*, and *Mafb* in podocytes, respectively. As the expression of several of these TFs was relatively low, possibly exacerbated further by transcript drop-outs, they did not show strong cell type enrichment. The regulon-based analysis, however, showed a very clear enrichment. SCENIC also successfully inferred multiple downstream target genes. The full list of regulons and their respective predicted target genes can be found in Supplementary Data 7, scaled and binarized regulon activity is available in Supplementary Data 8. Examples of regulon activity, corresponding TF expression, and predicted target gene expression are depicted in Fig. 2d and Supplemenatary Fig. 15a. For example, TFs such as *Eya1*, *Hoxc8*, *Hoxc9*, *Pax2*, *Spock2*, and *Wnt4* are important downstream targets within the regulon of nephron progenitor-specific TF *Uncx*, indicating an important transcriptional hierarchy of nephron development[36]. Corresponding snATAC-seq tracks for these predicted target genes along with the TF motif are depicted in Supplementary Fig. 15b. By comparing the number of cell type-specific TFs reported by HOMER and SCENIC to that from DEGs in RNA expression data, it became evident that our integrative *cis*-regulatory analysis with snATAC-seq and scRNA-seq datasets yielded significant benefits in discovering the TF-regulatory network over analyzing transcript data alone (Supplementary Fig. 16a, b).

**The regulatory trajectory of nephron progenitor differentiation**. All cells in the body differentiate from the same genetic template. Cell type-specific chromatin opening and closing events associated with TF binding changes set up the cell type-specific regulatory landscape resulting in cell type specification and development. We found that closing of open chromatin regions was the predominant event during the nephron progenitor differentiation (Supplementary Fig. 12d). We then evaluated the cellular differentiation trajectory in the snATAC-seq and scRNA-seq datasets (see "Methods" section). We identified the multiple nephron progenitor sub-groups (Fig. 3a, b), which will need to be carefully mapped to prior gene expression-driven and anatomical location-driven nephron progenitor sub-classification. Consistently, across

both data modalities, we identified that the podocyte precursors differentiated early from the nephron progenitor pool (Fig. 3a, b). The tubule cell trajectory was more complex with a shared intermediate stage and later differentiation into proximal tubules and distal tubules/loop of Henle (Fig. 3a, b and Supplementary Fig. 17a–c). We also integrated snATAC-seq and scRNA-seq data to obtain a single trajectory (see "Methods" section). The cell types in this dataset were correctly mapped and the trajectory resembled the path observed in individual analyses (Supplementary Fig. 17e–g). The robustness of developmental trajectories was further supported by RNA velocity analysis using Velocyto[37] (Supplementary Fig. 17d), and by comparing with previous human and mouse kidney developmental studies[9,13,14].

Building on both the SCENIC-generated gene regulatory network and the robust differentiation trajectories of the snATAC-seq and scRNA-seq datasets, we next aimed to understand chromatin dynamics, identify TFs and driver pathways for cell type specification. To this end, we first determined variation in chromatin accessibility along the three differentiation trajectories using ChromVAR[38]. ChromVAR estimates the accessibility dynamics of motifs in snATAC-seq data (see "Methods" section). The cell type-specific TF enrichment score matrix is shown in Supplementary Fig. 18a and Supplementary Data 9. We observed three different patterns when analyzing genes of interest (Fig. 3c): (1) Decrease of TF motif accessibility in all lineages. For example, *Sox11* motif enrichment score was high in nephron progenitor cells at the beginning of all three trajectories. It then decreased in all three lineages in parallel, underlining the role of *Sox11* in early kidney development. Several other TFs followed this pattern such as *Six2* and *Sox9*. (2) Cell type-specific maintenance of chromatin accessibility with advancing differentiation. We observed that chromatin accessibility for the *Wt1* motif was high initially but declined in proximal tubule and loop of Henle lineages, while its expression is increased in the podocyte lineage. This is consistent with the important role of *Wt1* in nephron progenitors and podocytes[39,40]. Other TFs that followed this pattern include *Foxc2* and *Foxl1*. (3) A de novo increase in chromatin accessibility with cell type commitment and advancing differentiation. For example, the chromatin accessibility of *Hnf4a* and *Pou3f3* motif increased in proximal tubule and loop of Henle trajectories, respectively, coinciding with the cellular differentiation program[41]. A large number of TFs followed this pattern such as *Mafb* (in podocytes), *Hnf4a* and *Hnf1a* (in proximal tubule), *Hnf1b* (in both proximal tubule and loop of Henle) as well as *Esrrb* and *Tfap2b* (in loop of Henle).

Next, we correlated changes in chromatin accessibility-based TF motif enrichment with TF expression, and their respective predicted target genes along the trajectories. To this end, we investigated TFs and target genes differentially expressed over scRNA-seq pseudotime (Supplementary Data 10). We noticed a good concordance of time-dependent changes of TF and predicted target gene expression along with motif enrichment, including the lineages for podocytes (e.g., *Foxc2*, *Foxl1*, *Mafb*, *Magi2*, *Nphs1*, *Nphs2*, *Plat*, *Synpo*, *Thsd7a*, *Wt1*, and *Zbtb7c*), proximal tubule (e.g., *Ace2*, *Atp1a1*, *Dab2*, *Hnf1a*, Hnf4a, *Hsd17b2*, *Lrp2*, *Maf*, *Slc12a3*, *Slc22a12*, *Slc34a1*, and *Wnt9b*), loop of Henle (e.g.,*Cyfip2*, *Cytip*, *Esrrb*, *Esrrg*, *Irx1*, *Irx2*, *Mecom*, *Pla2g4a*, *Pou3f3*, *Ppargc1a*, *Stat3*, *Sytl2*, *Tfap2b*, *Thsd4*, and *Umod*), as well as for both proximal tubule and loop of Henle (e.g., *Bhlhe40*, *Hnf1b*, and *Tmprss2*), respectively (Fig. 3c and Supplementary Fig. 18b). Most interestingly, we noticed two distinct patterns of how gene expression was related to chromatin accessibility. While gene expression of TFs increased over pseudotime, its corresponding motif accessibility either increased in parallel (such as *Hnf4a* and *Pou3f3*) or was maintained in a lineage-specific manner (such as *Wt1*). This might indicate different regulatory mechanisms during differentiation.

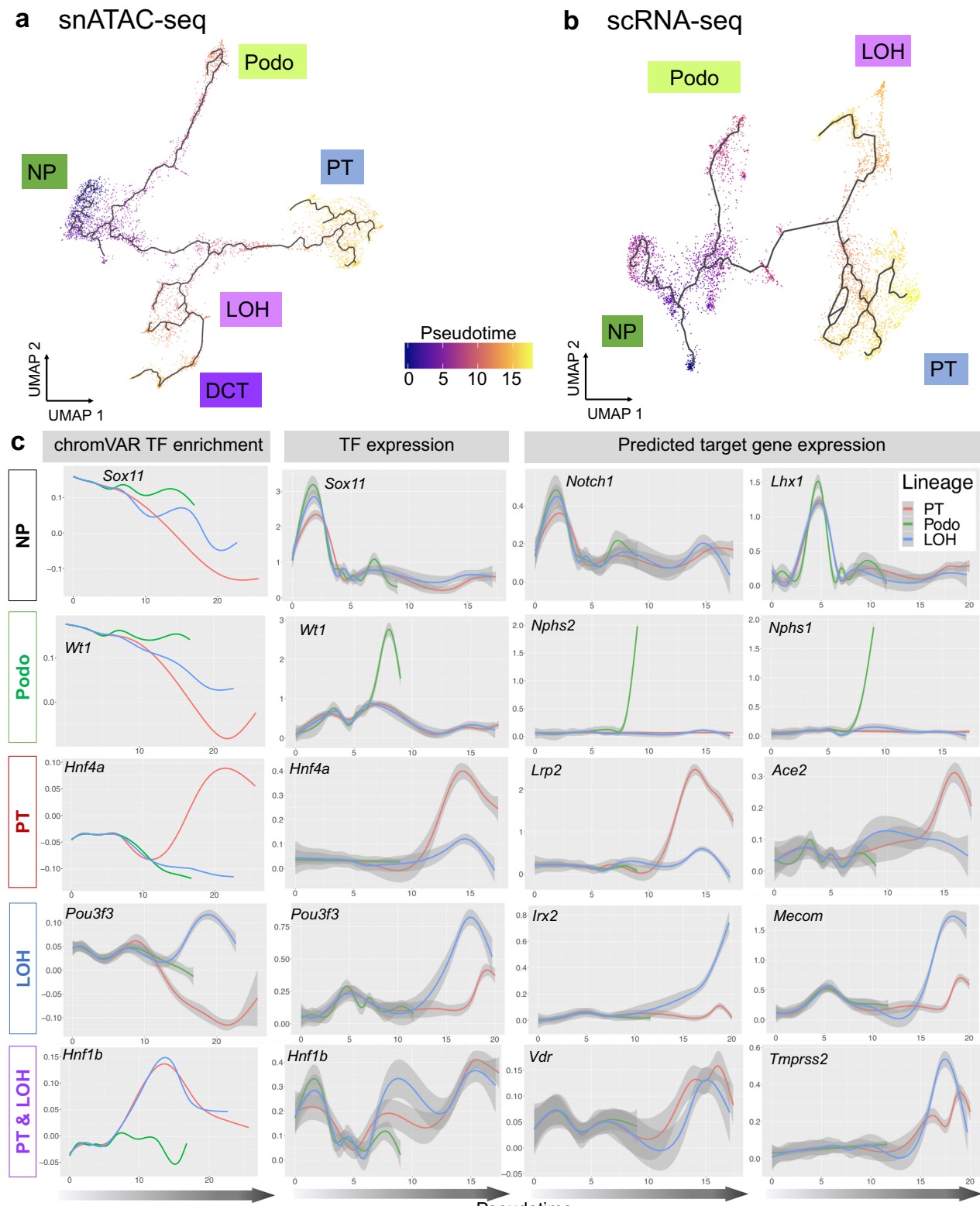

**Fig. 3 The cellular trajectory of nephron progenitor differentiation. a** UMAP representation of snATAC-seq nephron progenitor differentiation trajectory towards podocytes, proximal tubule, loop of Henle and distal convoluted tubule, respectively, as inferred by Cicero. Cells are colored by pseudotime. **b** UMAP representation of scRNA-seq nephron progenitor differentiation trajectory towards podocytes, proximal tubule and loop of Henle, respectively, as inferred by Monocle3. Cells are colored by pseudotime. **c** Pseudotime-dependent chromatin accessibility and gene expression changes along the proximal tubule (red), podocyte (green), and loop of Henle (blue) cell lineages. The first column shows the dynamics of chromVAR TF enrichment score, the second column shows the dynamics of TF gene expression values and the third and fourth column represent the dynamics of SCENIC-reported target gene expression values of corresponding TFs, respectively. Error bars denote 95% confidence intervals of local polynomial regression fitting. Additional examples are given in Supplementary Fig. 18b.

We next aimed to interrogate the stage-dependent chromatin dynamics along the identified differentiation trajectory. The differentiation trajectory was binned into 15 developmental steps based on the lineage specification in pseudotime inference (Supplementary Fig. 17b, c). These stages were labeled as NP1-3 (nephron progenitor), IM1-2 (intermediate cells), Podo1-3 (podocytes), PT1-3 (proximal tubule), LOH1-2 (loop of Henle), and DCT (distal convoluted tubule). By studying gene activity score enrichment of prior cell marker-based annotations, we showed that NP3 matches with renal vesicle signatures, IM1-2 and Podo1 match with Comma-shaped and S-shaped body that committed to tubule and podocyte fates, respectively (Fig. 4b). To study the chromatin opening and closing, we conducted differential chromatin accessibility analysis between subsequent stages. To understand the biological processes controlled by the epigenetic changes, we examined the nearby genes and performed functional annotation of these peaks (see "Methods" section). We found that open chromatin profiles were relatively stable in the early precursor stages such as NP1 to NP3, with fewer than 70 DAPs identified (Supplementary Data 11 and Supplementary Fig. 19). The podocyte differentiation branch was associated with marked increase in the number of DAPs (796 DAPs between NP3 and Podo1). This mainly represented the closing of chromatin areas around nephron progenitor-specific genes such as *Osr1*, *Gdnf*, *Sall1*, *Pax2* and opening of areas around podocyte-specific genes and key TFs such as *Foxc2* and *Efnb2*, both of which are validated to be important for early podocyte differentiation[42,43]. At later stages, there was a strong increase in chromatin accessibility of actin filament-based processes and a significant decrease in *Notch1*, *Notch2*, and *Ctnnb1* in the podocyte lineages (Supplementary Data 12). The chromatin changes from NP3 to intermediate cells 1 (IM1) were mainly associated with closing of the chromatin around *Osr1* and opening around tubule cell-specific TFs, such as *Lhx1* and *Pax3* (Fig. 4). The decrease in *Six2* expression only occurred at the IM2 stage, at which we also observed an increase in tubule specification genes such as *Hnf1a*. Gene ontology results from the 820 up-regulated peaks between PT1 and IM2 showed enrichment associated with typical proximal tubule functions including sodium-dependent phosphate transport, maintenance of osmotic response in the loop of Henle and active sodium transport in the distal convoluted tubule (Supplementary Fig. 19, the full list can be found in Supplementary Data 11, 12, and 13).

In addition to analyzing changes along the trajectory, we also examined cell-fate bifurcation events by comparing two descendant lineages. We found that podocyte specification from nephron progenitors was associated with differential opening of *Foxl1*, *Zbt7c*, and *Smad2* in the podocyte lineage and *Lhx1*, *Sall1*, *Dll1*, *Jag1*, *Cxcr3*, and *Pax3* in the other lineage, respectively. While the role of several TFs has been established for podocyte specification, the expression of *Foxl1* has not been described in the kidney until now (Fig. 4). Our analysis pinpointed that four peaks in the vicinity of *Foxl1* were accessible only in podocyte lineage, which locate in +53,381 bp, +152,832 bp, +237,019 bp, and +268,550 bp of the *Foxl1* TSS, respectively. To confirm the expression of *Foxl1* in nephron progenitors and podocytes, we performed immunofluorescence studies on developing kidneys (E13.5, P0, and P6). Consistent with the computational analysis, we found strong expression of FOXL1 in nephron progenitors (E13.5). At later stages, FOXL1 expression in glomerular podocyte was confirmed by co-localization with WT1 (Supplementary Fig. 20). Early and late stage expression of *Foxl1* were also confirmed in scRNA-seq data (Supplementary Fig. 21). While further experimental validation will be important, our study has illustrated the critical role of open chromatin state information and dynamics in cellular differentiation.

The intermediate cells (IM) gave rise to proximal and distal branches representing the proximal tubules, and the loop of Henle as well as distal convoluted tubule segments. The proximal tubule region was characterized by chromatin opening around *Hnf4a*, *Maf*, *Tprkb*, and *Gpat2*. The loop of Henle and distal convoluted tubule segments were remarkable for multiple DAPs in the vicinity of *Tfap2a*, *Tfap2b*, *Cited4*, *Ephb2*, *Ephb3*, *Hoxd8*, *Mecom*, and *Prmd16*, indicating a critical role for these TFs in distal tubule differentiation (Figs. 3c and 4 and Supplementary Fig. 21). Consistently, we saw a reduction in chromatin accessibility of *Six2* promoters and enhancers along all three trajectories (podocyte, proximal tubule, and loop of Henle) (Supplementary Fig. 22). There was also a decrease in accessibility of *Jag1* and *Heyl* in the distal loop of Henle segment, concordant with the putative role of Notch driving the proximal tubule fate[44] (Supplementary Data 14). Another striking observation was that tubule segmentation and specification occurred early by an increase in chromatin accessibility around *Lhx1*, *Hnf1a* and *Hnf4a* and *Maf* for proximal tubule and *Tfap2b* for loop of Henle. Terminal differentiation of proximal tubule and loop of Henle cells was strongly linked to nuclear receptors that regulate metabolism, such as *Esrra* and *Ppara* in proximal tubules and *Esrra* and *Ppargc1a* in the loop of Henle segment, once more indicating the critical role of metabolism of driving gene expression and differentiation[45].

**Stromal-to-epithelial communication is critical in the developing and adult kidneys.** Previous studies indicated that the survival, renewal, and differentiation of nephron progenitors is largely regulated through its cross-talk with the adjacent ureteric bud[46]. To investigate the complex cellular communication network, we used CellPhoneDB[47] to systematically infer potential cell–cell communication in the developing and adult kidney. CellPhoneDB provides a comprehensive database and a statistical method for the identification of ligand–receptor interactions in scRNA-seq data. Analysis of our scRNA-seq dataset indicated that the number of cell–cell interaction pairs was larger in developing kidney compared to the adult kidney (Fig. 5a). In the developing kidney, the stroma showed the greatest number of interactions among all cell types, coinciding with the well-known role of epithelial–stromal interactions in driving kidney development[44]. Of the identified interactions, many were related to stroma-secreted molecules such as collagen 1, 3, 4, 6, and 14 (Fig. 5b). Interestingly, the nephron progenitor cluster showed important ligand-receptor interaction between *Fgf1*, *Fgf8* as well as *Fgf9* and the corresponding receptor *Fgfr1*, which is consistent with the well-known role of FGF signaling in kidney development[48]. Of the manifold identified interactions in the fetal kidney, stromal interaction and the VEGF-involving interaction remained significant in the adult data set, underscoring the importance of endothelial-to-epithelial communication.

We next individually examined the expression of several key pathways known to play important roles in kidney development, such as Gdnf-Ret, sonic hedgehog, FGF, Bmp, Wnt, and others[9]. Expression of these key ligand–receptor pairs showed strong cell type specificity (Fig. 5c). For example, *Robo2* of the Gdnf-Ret pathway was expressed in nephron progenitors and in podocytes of P0 and adult kidney[49–51]. *Eya1*, however, is genetically upstream of *Gdnf* and acts as a positive regulator for its activation[52]. Consistently, we noted distinct cell type specificity of *Eya1* expression only in nephron progenitors, which was also true for other important signaling molecules such as *Ptch1*, *Smo*, and *Gli3* of the sonic hedgehog pathway. *Fgfr1* showed the highest expression in nephron progenitors as well as in fetal and adult stroma, underscoring the importance of FGF signaling for

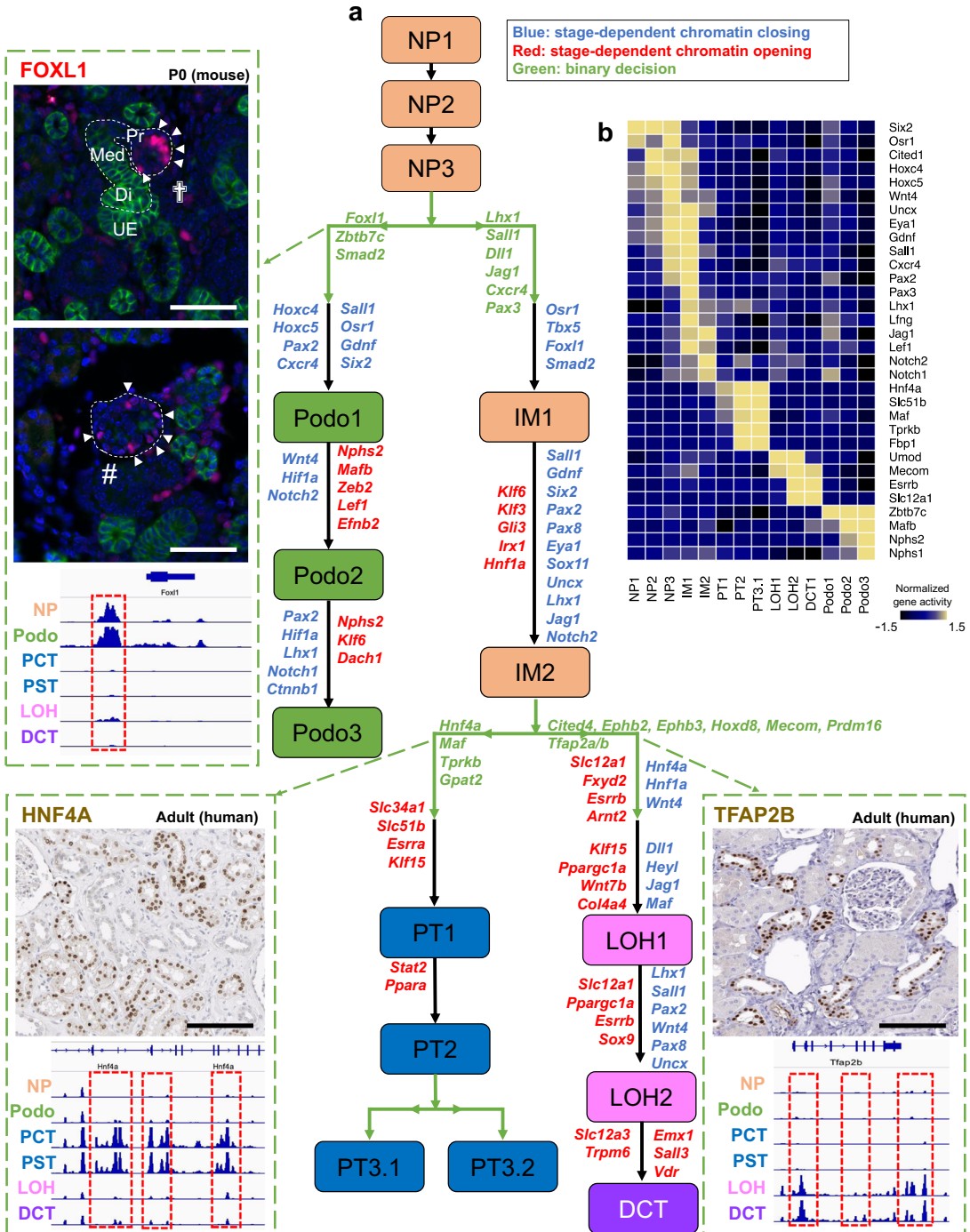

**Fig. 4 Chromatin dynamics of nephron progenitor differentiation. a** Di-graph representing cell type and lineage divergence, as derived from Cicero trajectory inference. Nephron progenitors (NP), podocytes (Podo), intermediate stage (IM), proximal tubule (PT), loop of Henle (LOH), and distal convoluted tubule (DCT) are connected with their developmental precursor stages and represented by ascending numbering. Arrows represent cell differentiation along respective trajectories. Genes listed next to the trajectories were derived from analyzing gene enrichment of differentially assessible peaks (DAPs) between two stages. Genes colored red were derived from the opening DAPs between two stages, genes colored blue were derived from the closing DAPs between two stages, and genes colored green were derived from opening DAPs between two branches. Three important genes, *Foxl1*, *Hnf4a*, and *Tfap2b* are shown along with their cell type-specific accessibility peaks and immunostaining results. Peaks that were open during the development of specific cell types are shown in red boxes. Immunofluorescence staining of P0 mouse kidney shows FOXL1 in red in the proximal part (Pr) of late S-shaped bodies (cross) and in podocytes within primitive glomeruli (#); green staining denotes E-Cadherin, blue DAPI; Med, medial part of S-shaped body; Di, distal part of S-shaped body; UE, ureteric epithelium; images are representative of three independent experiments, scale bars = 25 μm. HNF4A and TFAP2B in human adult kidney samples (images CAB019417 and HPA034683, respectively, taken from the Human Protein Atlas version 19.3, http://www.proteinatlas.org[83], https://creativecommons.org/licenses/by-sa/3.0/) are visualized by immunohistochemistry in brown, scale bar = 25 μm. In addition, gene expression changes along the three trajectories of genes that were identified by GREAT analysis to be nearby chromatin closing and opening events demonstrated to be consistent to chromatin information, examples are visualized in the top right subpanel. **b** Heatmap showing normalized gene activity scores of lineage marker genes in the 15 developmental stages.

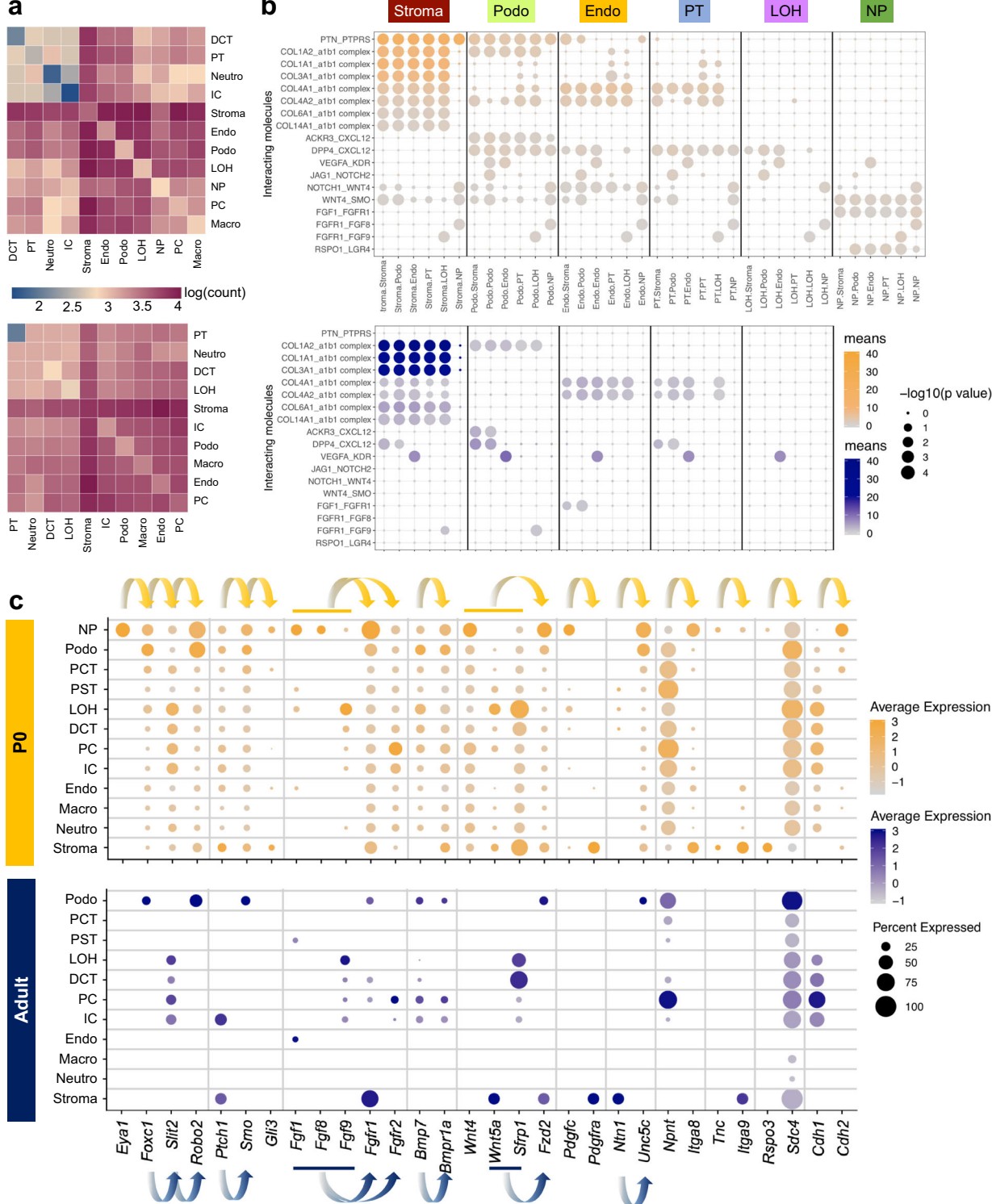

**Fig. 5 Cell–cell communication analysis in the developing and adult mice highlighted the critical role of stroma in driving cell differentiation.**
**a** Heatmaps showing the number of cell-cell interactions in the scRNA-seq dataset of P0 (top) and adult (bottom) kidneys, as inferred by CellPhoneDB. Dark blue and dark red colors denote low and high numbers of cell–cell interactions, respectively. **b** CellPhoneDB-derived measures of cell–cell interaction scores and *p* values. Each row shows a ligand-receptor pair, and each column shows the two interacting cell types, which is binned by cell type. Columns are sub-ordered by first interacting cell type into stroma, podocytes, endothelial cells, proximal tubule, loop of Henle, and nephron progenitors. Color scale denotes the mean values for all the interacting partners, where mean value refers to the total mean of the individual partner average expression values in the interacting cell type pairs. Orange scale denotes P0, blue scale denotes adult. Dot size denotes corresponding *p* values of the (one-sided) permutation test with 1000 permutations. **c** Dot plots of RNA expression of important cell–cell communication candidates within the Gdnf-Ret, Sonic hedgehog, Fgf, Bmp, Wnt, and other pathways in both P0 (top) and adult (bottom) kidney. Dot size denotes percentage of cells expressing the marker. Color scale represents average gene expression values, orange denotes P0, blue denotes adult. Arrows indicate ligand–receptor pairs.

cell–cell interactions in both the developing and developed kidney. Because not much is known about some of these markers, the significance of these putative interactions requires further investigation. For example, *Rspo3* has been implicated in nephron progenitor-associated interactions during nephrogenesis[10]. Mutations in the *Itga8* gene are known to cause isolated congenital anomalies of kidney and urinary tract in humans[50].

**Murine single cell chromatin accessibility implicates potential human kidney GWAS mechanisms, target regulatory regions, genes, and cell types**. Finally, we examined whether single cell level chromatin accessibility data can help implicate cell and gene targets for human kidney diseases. GWAS have been exceedingly successful in identifying nucleotide variations associated with specific diseases or traits. However, the majority of the identified genetic variants are in the non-coding region of the genome. Initial epigenome annotation studies indicated that GWAS hits are enriched in tissue-specific enhancer regions. As there are many different cell types in the kidney with differing function, understanding the true cell type specificity of these enhancers is critically important.

Here, we reasoned that murine single cell accessible chromatin information could be useful in nominating potential cell type-specific regulatory regions, and thereby the target cell type for the GWAS hits. We combined three recent kidney disease GWAS[53–55], and obtained 26,637 single nucleotide polymorphisms (SNPs) that passed genome-wide significance level. After lift-over, we identified 7923 variants by orthologous mapping from human to mouse.

First, we examined eGFR GWAS loci where functional validation studies reported conflicting results on target cell types and target genes (Fig. 6). First, we examined the region around *Uncx*, for which reproducible association with kidney function was shown in multiple kidney function GWAS[53,54]. Interestingly, the orthologous GWAS locus demonstrated a strong open chromatin region in nephron progenitors but not in any other differentiated cell types of the murine (Fig. 6a) or human (Fig. 6b) kidney. Consistently, by examine epigenome data (including H3K27ac, H3K4me1, H3K4me3, and ChIP-seq data across multiple stages), we observed enrichment for H3K27ac, H3Kme1, and *Six2* binding at this locus. We also detected *Uncx* gene expression in the fetal kidney samples, but not in the adult kidney (Fig. 6c).

Next, we analyzed the chromosome 15 GWAS region around *Dab2*, where we identified open chromatin regions in multiple kidney cells. *Dab2* expression, on the other hand, strongly correlated with open distal enhancer regions in proximal tubule cells (Fig. 6d). More importantly, this region was also accessible in human proximal tubule cells (Fig. 6e). This is consistent with earlier publications indicating the role of proximal tubule-specific *DAB2* expression in kidney disease development[56]. Interestingly, while single cell analysis indicated an additional distal enhancer in intercalated cells, the GWAS-significant region coincided with the proximal tubule-specific enhancer region and the open chromatin areas showed strong coregulation (Fig. 6f). Bulk kidney (ChIP-Seq) regulatory annotation indicated enhancer mark enrichment in the adult but not in the fetal kidneys.

We also examined loci where functional validation studies reported conflicting results on target cell types and target genes. The *SHROOM3* locus has shown a reproducible association with kidney function in multiple GWAS[55]. While one study indicated that the genetic variants were associated with an increase in *SHROOM3* levels in tubule cells inducing kidney fibrosis[57], the other suggested that the variant was associated with lower *Shroom3* levels in podocytes resulting in chronic kidney disease development[58]. We noticed that the expression of *Shroom3* in the

adult kidney was low (Fig. 6g). We observed open chromatin areas in multiple cell types including nephron progenitors, podocytes, and proximal tubule (Fig. 6g). The human kidney cell type-specific open chromatin landscape showed consistent overlap with multiple SNPs in multiple cell types (Fig. 6h). Interesting to note that one SNP with strong nephron progenitor-specific enrichment also coincided with the *Six2* binding area (Fig. 6i). While this finding is enticing, further experimental validation is warranted in order to uncover mechanistic insights. Closer views of all three loci are shown in Supplementary Fig. 23.

Lastly, we also overlapped all disease-associated SNPs (after lift-over) with peaks in snATAC-seq data. The full table including nearest genes is provided in Supplementary Data 15. These results indicate that variants associated with kidney disease development are located in regions with murine cell type-specific and developmental stage-specific regulatory activity, and illustrate the potential power of snATAC-seq in nominating target genes and target cell types for GWAS variants.

## Discussion

In summary, here we present the first cellular resolution open chromatin map with both developing and adult mouse kidney. Using this dataset, we identified key cell type-specific regulatory networks for kidney cells, defined the cellular differentiation trajectory, characterized regulatory dynamics and identified key driving TFs for nephron development, especially for the terminal differentiation of epithelial cells. Furthermore, our results shed light on potential cell types and target genes for genetic variants associated with kidney disease.

By performing massively parallel single cell profiling of chromatin state, we were able to define the key regulatory logic for each kidney cell type by investigating *cis*-regulatory elements, motif enrichment, and TF-target gene co-expression. These analyses helped the identification of key TFs for different kidney cell types. TF identification is challenging in scRNA-seq data, since the expression of several cell type-specific TFs is low[59]. We showed that by extracting motif information, snATAC-seq data provided critical complementary information for TF identification.

In our dataset, we also observed that the single cell open chromatin atlas was able to define more distinct cell types even in the developing kidney compared to scRNA-seq analysis. Given the continuous nature of RNA expression, it has been exceedingly difficult to dissect specific cell types in the developing kidney[9,10,13]. In addition, it has been difficult to resolve the cell type origin of transcripts expressed at low levels in scRNA-seq data. However, in our snATAC-seq data, the low dimensional embedding showed distinct clusters. In addition, snATAC-seq data were able to capture the chromatin state irrespective of gene expression magnitude. There were several examples where accessible peaks were identified in specific cell types even for genes expressed at low levels, such as *Shroom3*.

Our studies revealed dynamic chromatin accessibility that tracks with renal cell differentiation. These states may reveal mechanisms governing the establishment of cell fate during development. We found a coherent pattern between gene expression and open chromatin information, where the nephron progenitors differentiated into two branches representing podocytes and tubule cells[60]. We found that podocyte commitment occurred earlier, while tubule differentiation was more complex. The podocyte specification correlated with maintained expression of *Foxc2* and *Foxl1*, as evidenced by immunofluorescence. While Foxc2 has been known to play a role in nephron progenitors and podocytes, this is the first description of Foxl1 in kidney and podocyte development. We confirmed the key role of Hnf4a in proximal tubules and identified transcriptional regulators, such as

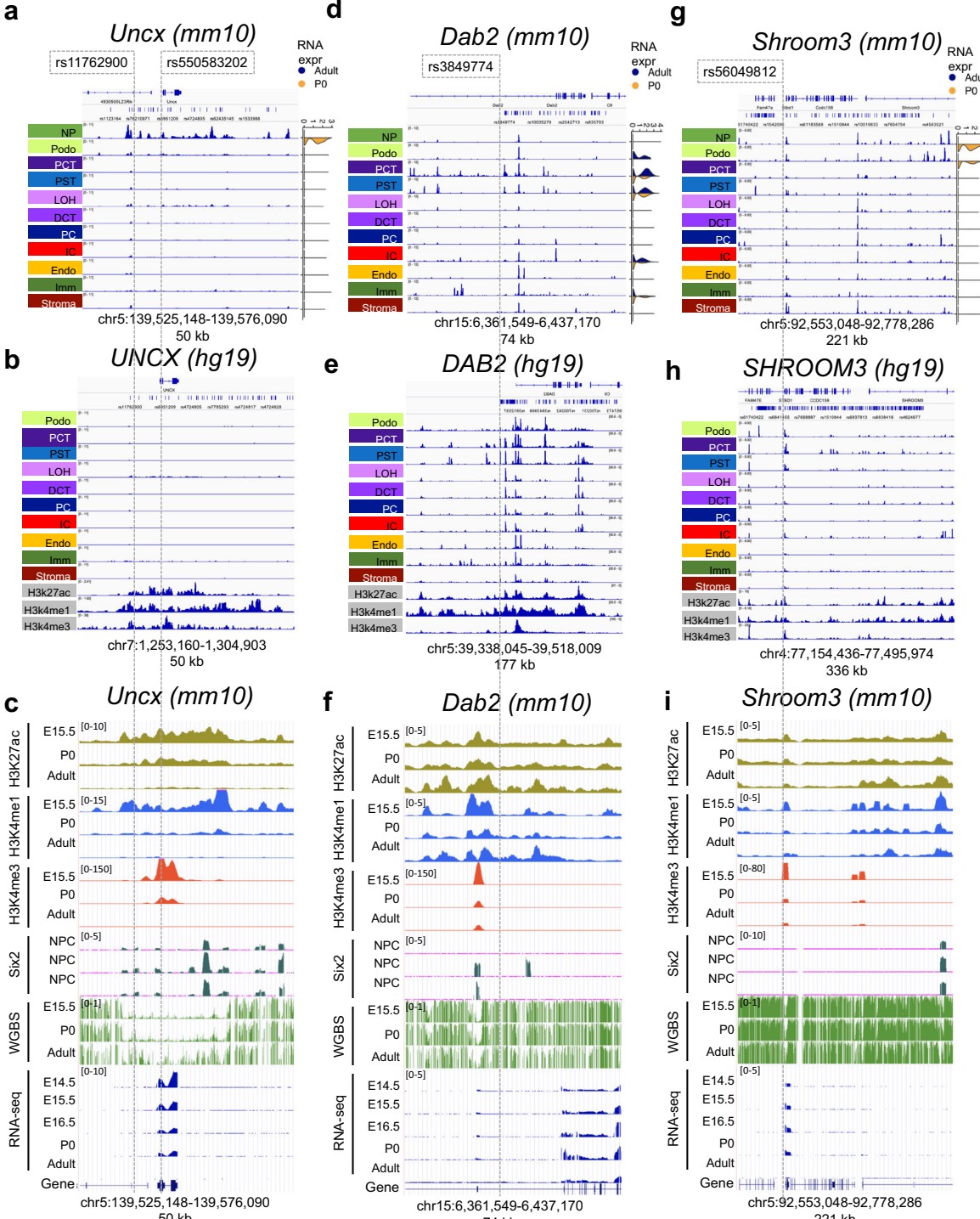

**Fig. 6 Single cell level chromatin accessibility highlighted human kidney GWAS target genes and cell types. a**, **d**, **g** Genome browser view of *Shroom3*, *Dab2*, and *Uncx* loci; from top to bottom: mouse orthologue of eGFR GWAS significant SNPs (after lift-over) mouse kidney single nuclei chromatin accessibility for nephron progenitors (NP), podocytes (Podo), proximal convoluted and straight tubules (PCT and PST), loop of Henle (LOH), distal convoluted tubule (DCT), collecting duct intercalated cells (IC), collecting duct principal cell types (PC), endothelial cells (Endo), immune cells (Immune) and stromal cells (Stroma). Data range in all tracks is set to the same scale. Examples of cell type-specific accessible chromatin overlapped with significant SNPs are highlighted with dashed lines. Right subpanel shows violin plots of cell type-specific gene expression in P0 (orange) and adult (blue) kidneys in the scRNA-seq dataset. **b**, **e**, **h** Corresponding genome browser views in adult human (hg19) kidney snATAC-seq data; from top to bottom: eGFR GWAS significant SNPs, adult human kidney single nuclei chromatin accessible landscape, whole kidney H3K27ac, H3K4me1, and H3K4me3 ChIP-seq tracks. The genomic location in human was matched to the mouse orthologue. Note that for *DAB2* (**f**), the image was mirrored to facilitate comparison along genomic read direction. Alternative genome browser views at more zoomed-in locations are available in Supplementary Fig. 23. **c**, **f**, **i** Whole mouse kidney epigenomics tracks from E14.5, E15.5, E16.5, P0, and adult mice. From top to bottom: H3K27ac, H3K4me1, H3K4me3, and Six2 ChIP-seq; whole genome bisulfate sequencing (WGBS); and bulk RNA-seq. The bottom Refseq visualization corresponds to the Refseq track at the top in **a**, **d**, **g**, respectively. Six2 binding signal in nephron progenitor cells.

Tfap2a for the distal portion of the nephron. Furthermore, the terminal differentiation of proximal tubule cells correlated with the increase in *Ppara* and *Esrra* expression, both of which are known regulators of oxidative phosphorylation and fatty acid oxidation[45]. Loop of Henle differentiation strongly correlated with *Essrb* and *Ppargc1a* expression. These studies potentially indicate that cell specification events occur early and metabolism controls terminal differentiation of tubule cells[61]. Impaired metabolic fitness of proximal tubules has been a key contributor to kidney dysfunction, explaining the critical association with tubule metabolism and function[62].

Furthermore, we show that murine single cell-level and differentiation stage-level epigenome annotation is potentially powerful for the annotation of human GWAS. Data from human kidney tissues are needed to understand whether the knowledge from mouse ortholog is transferrable to human, however, our analyses highlight the usefulness of murine analyses in the context of ethical and practicability issues in obtaining human samples, especially during developmental stages. Most identified GWAS signals are in the non-coding region of the genome and the target gene, as well as target cell type remain unknown. Our results indicate that multiple GWAS regions are conserved between mice and humans. Single cell open chromatin information enables not only to infer the implication of affected cell types, but also the understanding of co-regulation of the open chromatin area, which helps infer target genes. Here, we show that the GWAS variants mapped only to those regions where chromatin was open exclusively in nephron progenitors, whereas chromatin became inaccessible as differentiation progressed during later stages, such as for *Uncx*. This is an interesting and important potential mechanism, indicating that the altered expression of this gene might play a role in the development rewiring of the kidney. This mechanism would be similar to genes associated with autism that are known to be expressed in the fetal, but not in the adult stages[63], and highlights the critical role of understanding chromatin accessibility at multiple stages of differentiation.

While we have generated a large amount of high-quality data, this information will need further experimental validation. In addition, one needs to be aware of the limitations when interpreting different computational analyses, for example, motif enrichment analyses such as implemented by HOMER, SCENIC, and chromVAR are not able to distinguish between TFs with similar binding sites. Future high-throughput studies that analyze open chromatin and gene expression information from the same cells will be exceedingly helpful to correlate open chromatin, and gene expression information along the differentiation trajectory[31,64,65].

In summary, our dataset provides critical insight into the cell type-specific gene regulatory network, cell differentiation program, and human complex diseases-associated genes.

## Methods
**Material table**. Material table can be viewed in the Supplementary Data 16.

**Research animals**. Mice were housed in a temperature-controlled specific-pathogen-free facility under 12 h light/dark cycles (lights on at 07:00 h, off at 19:00 h). The animal experiments were reviewed and approved by the Institutional Animal Care and Use Committee (IACUC) of the University of Pennsylvania in accordance with the guidelines of the National Institutes of Health. All applicable international, national, and institutional guidelines for the care and use of animals were followed.

**Single cell RNA sequencing of P0 and adult mice**. One-day-old wild type mouse neonate and adult mouse (C57BL/6) kidneys were harvested and minced into 1 mm³ pieces and incubated with digestion solution containing Enzyme D, Enzyme R, and Enzyme A from Multi Tissue Dissociation Kit 1 (Miltenyi, 130-110-201) at 37 °C for 15 min with agitation. Reaction was deactivated by adding FBS to 10%, then solution was passed through a 40 μm cell strainer. After centrifugation at 500 × *g* for 5 min, cell pellet was incubated with 500 μL of RBC lysis buffer on ice for 3 min. We centrifuged the cells at 500 × *g* for 5 min at 4 °C and resuspended the

cells in 1X PBS for further steps. Cell number and viability were analyzed using Countess AutoCounter (Invitrogen, C10227). The cell concentration was 2.2 million cells/mL with 92% viability. Ten thousand cells were loaded into the Chromium Controller (10X Genomics, PN-120223) on a Chromium Single Cell B Chip (10X Genomics, PN-120262), and processed to generate single cell gel beads in the emulsion (GEM) according to the manufacturer's protocol (10X Genomics, CG000183). The library was generated using the Chromium Single Cell 3′ Reagent Kits v3 (10X Genomics, PN-1000092) and Chromium i7 Multiplex Kit (10X Genomics, PN-120262) according to the manufacturer's manual. Quality control for constructed library was performed by Agilent Bioanalyzer High Sensitivity DNA kit (Agilent Technologies, 5067-4626) for qualitative analysis. Quantification analysis was performed by Illumina Library Quantification Kit (KAPA Biosystems, KK4824). The library was sequenced on Illumina HiSeq 4000 system with 2 × 150 paired-end kits using the following read length: 28 bp Read1 for cell barcode and UMI, 8 bp I7 index for sample index and 91 bp Read2 for transcript.

**Single cell ATAC sequencing**. Kidneys from 3-week-old and 8-week-old mice were harvested, minced and lysed in 5 mL lysis buffer (10 mM Tris HCl pH 7.4, 10 mM NaCl, 3 mM MgCl₂, and 0.1% Nonidet™ P40 Substitute in nuclease-free water) for 15 min. Kidneys from P0 mice were minced and lysed in 2 mL lysis buffer for 15 min. Tissue lysis reaction was then blocked by adding 10 mL 1× PBS into each tube, and solution was passed through a 40 μm cell strainer. Cell debris and cytoplasmic contaminants were removed by Nuclei PURE Prep Nuclei Isolation Kit (Sigma, NUC-201) after centrifugation at 30,000 × *g* for 45 min. Nuclei concentration was calculated with Countess AutoCounter (Invitrogen, C10227). Diluted nuclei suspension was loaded and incubated in transposition mix from Chromium Single Cell ATAC Library & Gel Bead Kit (10X Genomics, PN-1000110) by targeting 10,000 nuclei recovery. GEMs were then captured on the Chromium Chip E (10X Genomics, PN-1000082) in the Chromium Controller according to the manufacturer's protocol (10X Genomics, CG000168). Libraries were generated using the Chromium Single Cell ATAC Library & Gel Bead Kit and Chromium i7 Multiplex Kit N (10X Genomics, PN-1000084) according to the manufacturer's manual. Quality control for constructed library was perform by Agilent Bioanalyzer High Sensitivity DNA kit. The library was sequenced on Illumina HiSeq 4000 system with 2 × 50 paired-end kits using the following read length: 50 bp Read1 for DNA fragments, 8 bp i7 index for sample index, 16 bp i5 index for cell barcodes and 50 bp Read2 for DNA fragments.

**Bulk ATAC sequencing**
*Nuclei preparation*. Bulk ATAC sequencing was performed with previously published methods[66,67]. Kidneys were minced and lysed in 5 mL lysis buffer (10 mM Tris HCl pH 7.4, 10 mM NaCl, 3 mM MgCl₂, and 0.1% Nonidet™ P40 Substitute in nuclease-free water) for 15 min. The reaction was then blocked by adding 10 mL 1× PBS into each tube, and solution was passed through a 40 μm cell strainer. The nuclei were centrifuged down at 500×*g* at 4 °C, resuspended in the resuspension buffer (10 mM pH = 7.5 Tris-HCl, 10 mM NaCl, 3 mM MgCl₂). Nuclei numbers were estimated with Countess AutoCounter (Invitrogen, C10227).

*Transposition*. Fifty thousand nuclei/sample were tagmented with Tagment DNA TDE1 Enzyme and Buffer Kit (Illumina, 20034198) in 50 μL reaction volume of transposition buffer (25 μL 2× TD buffer (Tagment DNA Buffer), 2.5 μL Tn5 transposase (Tagment DNA Enzyme 1), 0.5 μL 10% Tween-20, 0.5 μL 1% Digitonin, 16.5 μL 1× PBS, 5 μL nuclease-free water). The reaction was carried out at 37 °C for 30 min in a thermomixer.

*DNA purification and library construction*. Isolated DNA was purified by MinElute Reaction Cleanup Kit (Qiagen, 28204) by following the manufacturer's manual. The purified DNA was finally eluted in 10 μL elution buffer. The DNA was then amplified by NebNext High-Fidelity 2× PCR Master Mix (NEB, M0541S) and quantified by qPCR to make libraries. The libraries were purified by AMPure XP beads (Beckman Coulter, A63880). Quality control for constructed library was perform by Agilent Bioanalyzer High Sensitivity DNA kit. Libraries were submitted to 150 bp PE sequencing with Illumina HiSeq 3000 system.

**snATAC-seq data analysis**
*Data processing and quality control*. Raw fastq files were aligned to the mm10 (GRCm38) reference genome and quantified using Cell Ranger ATAC (v. 1.1.0). We only kept valid barcodes with number of fragments ranging from 1000 to 40,000 and mitochondria ratio less than 10%. One of the important indicators for ATAC-seq data quality is the fraction of peaks in promoter regions, so we did further filtration based on promoter ratio. We noticed the promoter ratio seemed to follow a binary distribution, with most of cells either having a promoter ratio around 5% (background) or more than 20% (valid cells) (Supplementary Fig. 2a). We therefore filtered out cells with a promoter ratio <20%. After this stringent quality control, we obtained 11,429 P0 single cells (5993 in P0_batch_1 and 5436 in P0_batch_2) and 16,887 adult single cells (7129 in P56_batch_3, 6397 in P56_batch_4, and 3361 in P21_batch_5).

*Preprocessing*. Since snATAC-seq data are very sparse, previous methods either conducted peak calling or binarization before clustering. Here, we chose to do binarization instead of peak calling for two reasons: 1) Peak calling is time consuming; 2) Many peaks are cell type-specific, open chromatin regions in rare populations are more likely to be treated as background. After binning fragments into 5 kb windows and removing the fragments not matched to chromosomes or aligned to the mitochondria, we binarized the cell–bin matrix. In order to keep only bins that were informative for clustering, we removed the top 5% most accessible bins and bins overlapping with ENCODE blacklist. The 484,606 remaining bins were used as input for clustering.

*Dimension reduction, batch effect correction, and clustering*. Clustering was conducted using snapATAC[19], a single-cell ATAC-seq algorithm scalable to large dataset. Previous benchmarking evaluation has shown that snapATAC was one of the best-performing methods for snATAC-seq clustering[68]. Diffusion map was applied as a dimension reduction method using function *runDiffusionMaps*. To remove batch effect, we used Harmony[20], in which the low dimensional embeddings obtained from the diffusion map were used as an input. Harmony iteratively pulled batch-specific centroid to cluster centroid until convergence to remove the variability across batches. After batch correction, a graph was constructed using k-Nearest Neighbor (kNN) algorithm with $k = 15$, which was then used as input for Louvain clustering. We used the first 20 dimensions for the Louvain algorithm. The number of dimensions was chosen using a method recommended by snapATAC, although we noticed that the clustering results were similar among a series of dimensions from 18 to 30.

*Cell type annotation*. We used a published list of marker genes[9,21] to annotate kidney cell types. In order to infer gene expression of each cell type, we built a cell-gene activity score matrix by integrating all fragments that overlapped with gene transcript. We used GENCODE Mouse release VM16[27] as reference annotation.

*Comparison with mouse ATAC Atlas dataset*. We compared our cell type annotations and QC procedures with previously published sciATAC-seq data[17] with mouse adult kidney samples included. As for the QC procedure, in the published mouse ATAC atlas, cells were kept with at least 500 UMIs and less than 10% mitochondria reads, while in our analysis, we kept cells with at least 1000 UMIs and less than 10% mitochondria reads. In addition, we also filtered out cells with promoter ratio below 20%. Despite the more stringent QC step, the median number of accessible peaks per cell is comparable, with 4413 in our dataset and 4333 in the Mouse ATAC Atlas dataset.

Our dataset contains larger number of cells (28,346 in our dataset vs. 6299 in the mouse atlas dataset), allowing us to improve the cell type resolution of the prior map. For example, we were able to identify immune and CNT cells and separate the PCT and PST segments of the proximal tubules and IC and PC cells in the collecting duct. Thus, our dataset includes 13 annotated cell types while the previous dataset has 6. In addition, the different resolution may also result in the difference in experimental technologies.

Among the annotated cells, the cell type proportion is mostly consistent. By investigating the cell type-specific marker genes reported in the Mouse ATAC Atlas dataset, we observe good consistency between the cell type-specific peaks, indicating a consistent cell type annotation.

*Peak calling and visualization*. Peak calling was conducted for each cell type separately using MACS2[22]. We aggregated all fragments obtained from the same cell types to build a pseudo-bulk ATAC data and conducted peak calling with parameters "–nomodel–keep-dup all–shift 100–ext 200–qval 1e−2 –B–SPMR–call-summits". By specifying "–SPMR", MACS2 generated "fragment pileup per million reads" pileup files, which were converted to bigwig format for visualization using UCSC bedGraphToBigWig tool.

We also visualized public chromatin ChIP-seq data and RNA-seq data obtained from ENCODE Encyclopedia with the following identifiers: ENCFF338WZP, ENCFF872MVE, ENCFF455HPY, ENCFF049LRQ, ENCFF179NTO, ENCFF071PID, ENCFF746MFH, ENCFF563LOO, ENCFF184AYF, ENCFF107NQP, ENCFF465THI, ENCFF769XWI, and ENCFF591DAX. The Six2 ChIP-seq data were obtained from the ref. [69] and the WGBS data were obtained from ref. [70].

*Genomic elements stratification*. Mouse mm10 genome annotation files were download from UCSC Table Browser (https://genome.ucsc.edu/cgi-bin/hgTables) using GENCODE VM23. TSS upstream 5 kb regions were included as promoter regions, but the results were similar when using 2 kb upstream regions as promoters. We then studied the number of overlapped regions between open chromatin regions identified from the snATAC-seq and bulk ATAC-seq dataset and genome annotations. Since one open chromatin region could overlap with multiple genomic elements, we defined an order of genomic elements as exon > 5′-UTR > 3′-UTR > intron > promoter > distal elements. To be more specific, if one peak overlapped with both exon and 5′-UTR, the algorithm would count it as an exon-region peak.

*Identification of differentially accessible regions*. Peaks identified in each cell type were combined to build a union peak set. Overlapping peaks were then merged to one peak using *reduce* function from the GenomicRanges package. This resulted in 300,755 peaks, which were used to build binarized cell-by-peak matrix. Differentially accessible peaks (DAPs) for each cell type were identified by pairwise peak comparison.

Because of the binary nature of the single cell peak matrix, Fisher's exact test has been widely used to compute differentially accessible peaks[65,71–73]. To define cell type-specific peaks, we required the peaks to be stringently expressed in one specific cell type. Specifically, for each peak we conducted a Fisher's exact test between a cell type and each of the other cell types. Peaks with corrected $p$ values (Benjamini–Hochberg approach) below significance level (0.05) in all pairwise tests were defined as cell type-specific peaks. This approach was inspired by our previous experience on scRNA-seq data differential expression analysis[21], and similar to another published snATAC-seq paper[73]. In total, we obtained 60,683 highly specific DAPs, which were used for motif enrichment analysis.

*Motif enrichment analysis*. Motif enrichment analysis was conducted using DAPs by HOMER[34] with hypergeometric test for enrichment (one-sided). We used the parameters background = "automatic" and scan.size = 300. We noticed that de novo motif identification only generated few significant results, so we focused on known motifs for our following study. We used the significance level of 0.05 for Benjamini–Hochberg (BH) corrected $p$-value to determine the enriched results. The motif enrichment results are provided in Supplementary Data 6.

*Peak–peak correlation analysis*. Peak–peak correlation analysis was conducted using Cicero[33]. In order to find developmental stage-specific peak-peak correlations, the analysis was conducted for P0 and adult separately. Cicero uses Graphic Lasso with distance penalty to assess the co-accessibility between different peaks. Cicero analysis was conducted using the *run_cicero* function with default parameters. A heuristic cutoff of 0.25 score of co-accessibility was used to determine the connections between two peaks.

*snATAC-seq trajectory analysis*. snATAC-seq trajectory was conducted using Cicero, which extended Monocle3 to the snATAC-seq analysis. We obtained the preprocessed P0 snATAC-seq cell-peak matrix from snapATAC as input for Cicero and conducted dimension reduction using latent semantic indexing (LSI) and visualized using UMAP. Trajectory graph was built using the function *learn_graph*. Batch effect was not observed between the two P0 batches, and the trajectory graph was consistent with cell type assignment with clustering analysis (Fig. S17a, b).

In order to study how open chromatin changes are associated with the cell fate decision, we first binned the cells into 15 groups based on their pseudotime and cell type assignment. Next, we studied the DAPs between each group and its ancestral group using the same methods described above. The number of newly open and closed chromatins were reported using pie charts. The exact peak locations are provided in the Supplementary Data 11.

*Genes and gene ontology terms associated with snATAC-seq trajectory*. Based on the binned trajectory graphs and DAPs between each group and its ancestral group, we next used GREAT tool v4.0.4[74] to study the enrichment of associated genes and gene ontology (GO) terms along the trajectory. We used the newly open or closed peaks as test regions and all the peaks from peak-calling output as the background regions for the analysis. The output can be found in the Supplementary Data 13 and 14.

*Predict cis-regulatory elements*. We implemented two methods to study *cis*-regulatory elements in the snATAC-seq data. The first method was inspired by Zhu et al.[30], which was based on the observation that there was co-enrichment in the genome between the snATAC-seq cell type-specific peaks and scRNA-seq cell type-specific genes. This method links a gene with a peak if (1) they were both specific in the same cell type, (2) they were in *cis*, meaning that the peak is in ±100 kb region of the TSS of the corresponding gene, and (3) the peak did not directly overlap with the TSS of the gene. This method successfully inferred several known distal elements such as for *Six2* and *Slc6a18* (Supplementary Fig. 13c).

Alternatively, we assessed the co-accessibility of two peaks. We implemented Cicero[33], which aggregates similar cells to obtain a set of "meta-cells" and address the issue with sparsity in the snATAC-seq data. We used *run_cicero* function with default parameters to predict *cis*-regulatory elements (CREs). Although it is recommended to use 0.25 as a cutoff for co-accessibility score, we noticed that this resulted in a great amount of CREs, which could contain many false positives. Thus, we used a more stringent score of 0.4 for the cutoff and retained 232,380 and 206,701 CRE links in the P0 and adult data, respectively.

**Bulk ATAC sequencing analysis**. Bulk ATAC-seq raw fastq files were processed using the end-to-end tool ENCODE ATAC-seq pipeline (Software and Algorithms). This tool provided a standard workflow for ATAC-seq data quality control, adapter removal, alignment, and peak calling. To obtain high quality ATAC-seq peaks, peak calling results from two biological replicates were compared and only those peaks that were present in both replicates were kept, which were further used to compare with snATAC-seq peaks.

**Correlation of bulk and single nuclei ATAC sequencing data**. snATAC-seq reads were aggregated to a pseudo-bulk data for the comparison purpose. To prevent the effect of sex chromosome and mitochondria chromosome, reads from chromosome X, Y and M were excluded from our analysis. We used multi-BigwigSummary tool from deeptools[75] to study the correlation between different samples. Specifically, the whole genome was binned into equally sized (10 kb) windows, and the reads in each bin were aggregated, generating a bin-read count vector for each of the sample. The correlation of these vectors was computed as a measure of pairwise similarity between samples.

To compare the number of peaks in these two datasets, we used as input the narrowpeak files from the snATAC-seq and bulk ATAC-seq analysis. We filtered out bulk ATAC-seq peaks with $q$ value > 0.01 to be consistent with the snATAC-seq setting. Since the snATAC peaks were called after merging different time points, we also took the union set of bulk ATAC-seq peaks from different time points. We then used *findoverlap* function in GenomicRanges package[76] to identify overlapping peaks.

**Comparison between single nuclei ATAC sequencing data and single cell RNA sequencing data**. In order to compare the cluster assignment between snATAC-seq data and scRNA-seq data, we obtained the average gene expression values and peak accessibility in each cluster for P0 and adult samples separately. We next transformed snATAC-seq data by summing up the reads within gene body and 2 kb upstream regions to build gene activity score matrix, as suggested in Seurat[23]. Then, we normalized the data and computed the mean expression and mean gene activity scores in each cell type, and calculated $z$-scores of each gene. Pearson's correlation coefficient was then calculated among top 3000 highly variable genes between snATAC-seq data and scRNA-seq data. We found high concordance between these two datasets in terms of cell type assignment (Fig. 1g and Supplementary Fig. 3b).

**Single cell RNA sequencing data analysis**

*Alignment and quality control*. Raw fastq files were aligned to the mm10 (Ensembl GRCm38.93) reference genome and quantified using CellRanger v3.1.0. Seurat R package v3.0 was used for data quality control, preprocessing, and dimensional reduction analysis. After gene-cell data matrix generation of both P0 and adult datasets, matrices were merged and poor-quality cells with <200 or >3000 expressed genes and mitochondrial gene percentages >50 were excluded, leaving 25,138 P0 and 18,498 adult cells for further analytical processing, respectively (Supplementary Fig. 4a, b).

*Pre-processing, batch effect correction, and dimension reduction*. Data were normalized by RPM following log transformation and 3000 highly variable genes were selected for scaling and principal component analysis (PCA). Harmony R package v1.0[20] was used to correct batch effects. The top 20 dimensions of Harmony embeddings were used for downstream uniform manifold approximation and projection (UMAP) visualization and clustering (Supplementary Fig. 4c, d).

*Cell clustering, identification of marker genes, and differentially expressed genes*. Louvain algorithm with resolution 0.4 was used to cluster cells, which resulted in 18 distinct cell clusters. A gene was considered to be differentially expressed if it was detected in at least 25% of one group and with at least 0.25 log fold change between two groups and the significant level of Benjamini–Hochberg (BH) adjusted $p$ value < 0.05 in Wilcoxon rank sum test was used. We used a list of marker genes[9,21] to manually annotate cell types. Two distal convoluted tubule clusters were merged based on the marker gene expression, resulting in a total of 17 clusters (Supplementary Fig. 5a–c).

*Ambient RNA quantification*. As in droplet based scRNA-seq experiments, there is always a certain amount of background mRNAs present that get distributed into droplets and sequenced along with cells[77]. In order to quantify the net effect of ambient RNA contamination, we used R package SoupX[24]. Function *autoEstCount* was used to estimate the contamination fraction in P0 and adult batches separately. We visualized the change in expression due to soup correction in UMAP space. Function *adjustCounts* was used to correct the count expression matrices for downstream processing. We then used the corrected matrices and reran the whole Seurat pipeline with the same parameters. The average expression of genes per cluster were used for Pearson correlation coefficient analysis to compare between matrices with and without ambient RNA correction. As results with and without ambient RNA were similar (Supplementary Figs. 6 and 7), results without ambient RNA correction are shown throughout the manuscript.

*Subclustering of stroma populations*. To investigate the heterogeneity within the stroma clusters, we conducted subclustering analysis on P0 (Supplementary Fig. 9) and adult (Supplementary Fig. 10) stroma cells. Using recently reported marker genes, we were able to recapitulate some cell types including mesangial cells, fibroblasts, smooth muscle cells, and juxtaglomerular cells. We also found some subpopulations with various signatures. Further experimental validation is needed to determine whether these populations are due to artifacts in single cell data (doublets or contaminated reads) or real populations.

*scRNA-seq trajectory analysis*
Monocle3
To construct single cell pseudotime trajectory and to identify genes whose expression changed as the cells underwent transition, Monocle3 v0.1.3[78] was applied to P0 cells of the following Seurat cell clusters: nephron progenitors (NP), proliferating cells, stroma-like cells, podocytes, loop of Henle (LOH), early proximal tubule (PT), proximal convoluted tubule (PCT), and proximal straight tubule (PST) cells.
To show cell trajectories from both small (nephron progenitors) and large cell populations (proximal tubule), an equal number of 450 cells per cluster was randomly sub-sampled. Cells were re-clustered by Monocle3 using a resolution of 0.0005 with kNN $k = 29$. Highly variable genes along pseudotime were identified using *differentialGeneTest* function and cells were ordered along pseudotime trajectory. NP cluster was defined as earliest principal node. In order to find genes differentially expressed along pseudotime, trajectories for podocytes, loop of Henle, and proximal tubule clusters were analyzed separately with the *fit_models* function of Monocle3. Genes with a $q$ value < 0.05 in the *differentialGeneTest* analysis were kept. In an alternate approach, *graph_test* function of Monocle3 was used and trajectory-variable genes were collected into modules at a resolution of 0.01.

RNA velocity
To calculate RNA velocity, Python-based Velocyto command-line tool as well as Velocyto.R package were used as instructed[37]. We used Velocyto to calculate the single-cell trajectory/directionality using spliced and unspliced reads. From loom files produced by the command-line tool, we subset the exact same cells that were previously selected randomly for Monocle trajectory analysis. This subset was loaded into R using the SeuratWrappers v0.1.0 package. RNA velocity was estimated using gene-relative model with kNN cell pooling ($k = 25$). The parameter $n$ was set at 200, when visualizing RNA velocity on the UMAP embedding.

*Gene regulatory network inference*. In order to identify TFs and characterize cell states, we employed *cis*-regulatory analysis using the R package SCENIC v1.1.2.2[35], which infers the gene regulatory network based on co-expression and DNA motif analysis. The network activity is then analyzed in each cell to identify recurrent cellular states. In short, TFs were identified using GENIE3 and compiled into modules (regulons), which were subsequently subjected to *cis*-regulatory motif analysis using RcisTarget with two gene-motif rankings: 10 kb around the TSS and 500 bp upstream. Regulon activity in every cell was then scored using AUCell. Finally, binarized regulon activity was projected onto Monocle3-created UMAP trajectories.

*Ligand–receptor interactions*. To assess cellular crosstalk between different cell types, we used the CellPhoneDB repository to infer cell–cell communication networks from single cell transcriptome data[47]. We used the Python package Cell-PhoneDB v2.1.2 with the database v2.0.0 to predict cell type-specific ligand–receptor interactions as per the authors' instructions. Only receptors and ligands expressed in more than 5% of the cells in the specific cluster were considered. Thousand iterations of permutation test were conducted and $p$ values were corrected using FDR methods.

**Human kidney tissue processing**. Ten adult human kidney samples were obtained from the non-tumor tissue of six partial or radical nephrectomy patients from the Hospital of the University of Pennsylvania. Institutional Review Boards at the University of Pennsylvania reviewed this study. This project utilized de-identified kidney biospecimens collected via CHTN (Cooperative Human Tissue Network), and therefore was considered "exempt" by the local IRB. The work was completed in compliance with all relevant ethical regulations. Fresh tissues were shipped to the lab in RPMI on ice the same day after nephrectomy. Nuclei preparation, library construction, and sequencing methods were the same as for mouse kidneys described above.

**snATAC-seq data analysis of human kidneys**

*Data processing and quality control*. Raw fastq files were aligned to the b37 (GRCh37) genome, quantified using Cell Ranger ATAC (v. 1.2.0). The quality control and barcodes filtration steps were similar as mice samples described above. Briefly, we only keep cells with promoter ratio between 20 and 60%, UMIs between $10^3$ and $10^5$, and mitochondria ratio below 10%. In total, we obtained 61,440 single cells across ten adult human kidney samples after quality control.

*Preprocessing, dimension reduction, batch effect correction, clustering, and cell type annotation*. Filtered barcode-by-cell matrices were processed with SnapATAC pipeline similar to that in mouse samples described above. Specifically, we used 5 kb bin size to create cell-by-bin matrices, following dimension reduction, we removed batch effect by Harmony. We used $k = 15$ for the k-nearest neighbor algorithm and used the first 24 dimensions and resolution = 0.8 for clustering with the Louvain algorithm.

*Peak calling and visualization*. We used MACS2 for peak calling of each cell type with the same parameters as for mouse samples described above. To visualize the

chromatin ChIP-seq data for H3K4me3, H3K4me1, and H3K27ac, we used public data from GEO with the following identifiers: GSM1586397, GSM3716711, and GSM3716714[79].

**Immunofluorescence staining**. Mouse kidneys were fixed with 4% paraformaldehyde overnight, rinsed in PBS, and dehydrated for paraffin embedding. Antigen retrieval was performed using Tris-EDTA buffer pH 9.0 with a pressure cooker (PickCell Laboratories, Agoura Hills, CA) and antibody staining performed as described[80]. Antibodies used were as follows: guinea pig FOXL1 (1:1500)[81], mouse E-cadherin (1:250; BD Transducton 610182, Franklin Lakes, NJ), goat WT1 (clone F6) (1:50; Santa Cruz sc-7385). Cy2-conjugated, Cy3-conjugated, and Cy5-conjugated donkey secondary antibodies (1:2000) were purchased from Jackson ImmunoResearch Laboratories, Inc, AlexaFluor 488-conjugated donkey secondary antibodies were from LifeSciences (1:1000). Fluorescence images were collected on a Keyence microscope.

**Statistical information**. The differential accessible analysis was conducted by pairwise Fisher's exact test. The differential expression analysis was conducted using Wilcoxon rank sum test. The motif enrichment was based on hypergeometric test. All statistical tests were multitest corrected using FDR method and a significance level of 0.05 was used throughout the manuscript.

**Reporting summary**. Further information on research design is available in the Nature Research Reporting Summary linked to this article.

## Data availability
Raw data, processed data, and metadata from mouse samples have been deposited in GEO with the accession codes GSE157079. The processed data and metadata can also be viewed, analyzed, and downloaded at susztaklab.com/developing_adult_kidney/snATAC/, susztaklab.com/developing_adult_kidney/scRNA/, and susztaklab.com/developing_adult_kidney/igv/. The raw data, processed data, and metadata from human samples are available at https://www.diabetesepigenome.org. In this study, we downloaded public data from the following database with accession numbers: GUDMAP (RID:Q-Y4CY); ENCODE (ENCFF338WZP, ENCFF872MVE, ENCFF455HPY, ENCFF049LRQ, ENCFF179NTO, ENCFF071PID, ENCFF746MFH, ENCFF563LOO, ENCFF184AYF, ENCFF107NQP, ENCFF465THI, ENCFF769XWI, ENCFF591DAX); GEO (GSM1051156, GSM3716711, GSM3716714, GSM1586397). Further information and requests for resources and reagents should be directed to and will be fulfilled by the lead contact: Katalin Susztak. email: ksusztak@pennmedicine.upenn.edu.

## Code availability
Codes are available at https://github.com/Zhen-Miao/dev-kidney-snATAC. (https://doi.org/10.5281/zenodo.4421623)[82].

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

## Acknowledgements

This work in the Susztak lab is supported by the NIH DK076077, DK087635 and DK105821. M.S.B. is supported by a German Research Foundation grant (BA 6205/2-1). We thank the University of Pennsylvania Diabetes Research Center (DRC) for the use of the next generation sequencing Core (P30-DK19525). We thank Dr. Pablo G. Cámara, Dr. Wenliang Wang, Qin Zhu, Feiya Ou, Siling Du, and Lulu Liu for constructive suggestions that improve the manuscript.

## Author contributions

Z.M. and K.S. designed and conceived the experiment. Z.Y.M., J.W., R.S., and T.A. conducted the experiment. Z.M. conducted snATAC-seq bioinformatics analysis with advice from K.S., H.L., M.S.B., and J.K. M.S.B. and Z.M. conducted scRNA-seq bioinformatics analysis with advice from K.S., M.L., J.K., and M.P. M.S.B., A.M.K., and A.Y.K. conducted immunofluorescence staining with supervision from K.H.K. All authors discussed and commented on the results. K.S., Z.M., and M.S.B. wrote the manuscript and all authors edited and approved of the final manuscript.

## Competing interests

The authors declare no competing interests.
