## [Peer Review File · Nature Communications]

REVIEWER COMMENTS

Reviewer #1 (Remarks to the Author):

The manuscript from Professor Susztak's group contains some really extensive and important data on the epigenetic underpinnings of cell-specific developmental programmes in the murine kidney. The manuscript will be of interest to biologists, developmental biologists, nephrologists and those interested in the applications of omics technologies. The paper is beautifully written and the interpretation of the data is generally careful. However, I think that the sections pertaining to the integration of data from human genome-wide association studies with the outputs from chromatin accessibility and epigenetic marks from the mouse kidney require some clarifications.

1. Zooming on three specific GWAS regions overlapping with SHROOM3, Dab2, Uncx is well justified. What I feel requires a more explicit confirmation is that the authors' analysis provides insights into mouse orthologs of human genes overlapping with GWAS CKD signals. The "ortholog" term is used only once in the legend to Figure 6.

2. Epigenetic marks of active transcription and gene enhancers in mouse kidney are nicely illustrated in Panel B of Figure 6. These marks should be now available for human kidneys in RoadMap Epigenomics. I wonder if this would be helpful to integrate them in Figure 6?

3. I feel that the discussion should make a much stronger point on the need of using human kidney cells and tissues for the purpose of deciphering the biological mechanisms underlying signals of association with CKD in GWAS. The authors rightly state in Discussion the single cell transcriptome and epigenome data are necessary to understand the findings from GWAS because their molecular underpinnings are so tissue-, cell- and context specific. It is unlikely that these highly sophisticated regulatory mechanisms would be conserved between the rodent and the human. The existence of an ortholog to a potential human CKD GWAS eGene in a mouse does not necessarily mean that the identified murine epigenetic and transcriptional mechanisms can be translated to the human.

4. Line 98 in introduction - I would not immediately dismiss the utility of "bulk tissue" in this kind of analysis given that the new bio-informatic pipelines taking advantage of the existing human cell data can adjust for potential heterogeneity in cell compositions between the samples.

5. I think that the statements pertaining to eQTL analysis in human bulk tissues have to be more careful – these studies have been extremely useful in identifying kidney eGenes – the first step to understand molecular mechanisms of CKD GWAS (i.e. Nat Med. 2018;24(11):1721-1731; Nat Commun. 2018;9(1):4800.). Lines 600-601: I am not clear how single cell sequencing can overcome the linkage disequilibrium problem in the eQTL analysis.

6. Figure 6 is very nice already but I think it would benefit from a little more specific description in the legend for all layers (i.e. SNPs, etc).

Reviewer #2 (Remarks to the Author):

In the manuscript entitled "Single cell resolution regulatory landscape of the mouse kidney highlights cellular differentiation programs and renal disease targets", Miao and colleagues present a large resource of single cell ATAC-seq data of mouse kidneys at different ages collected on the 10X Genomics platform. Alongside single cell chromatin data, the authors also generate single cell RNA-seq data in the same time points. My expertise is in single cell genomics, not kidney biology, and so I present a critique of the methods here. This work presents a unique opportunity to understand some of the dynamics of kidney development that happen postnatally in the mouse. The authors identify relevant cell types, uncover master regulators (both known and novel), chart the likely dynamics of cell type differentiation from progenitors, and identify specific regulatory elements that are putatively important in kidney development and disease. This work is novel, and I believe it will be of general interest to the field. However, there are several issues that need to be addressed. In the main, several of the analyses should be refined or removed, and some of the relevant literature should be better referenced and compared. My major and minor criticism are presented below:

Major criticism

1) Single cell ATAC-seq data on the kidney already exists (either published or preprint) and is

uncited in the manuscript. <https://doi.org/10.1016/j.cell.2018.06.052> is a single cell ATAC-seq atlas of the adult mouse, including kidney. <https://doi.org/10.1101/865931> is a preprint that includes mouse kidney single cell ATAC-seq data. The work by Miao et al. includes time points not previously profiled, but the authors should reference the other work and compare their results more directly. It would be interesting to see if the studies cluster together well and if the relative proportions of cell types are similar. Direct comparison of QC measures would also be informative.

2) A major strength of the submitted manuscript is the inclusion of time points, which allows us to observe the postnatal development of the tissue, but the study should include more time points. Capturing the correct intermediate states depends critically on capturing appropriate intermediate time points. At a minimum, the authors should justify why these three specific time points were chosen and make clear why others are not needed.

3) The analysis of Cicero link distances is inappropriate given the underlying model. In the section on lines 242-250, the authors discuss an analysis of the distance between co-accessible peaks. They make the argument that peaks up to 500kb show strong co-accessibility scores. While it's true that there are outliers with strong measures even at this distance, the general trend is of decreasing scores with increasing distance. This happens because the Cicero model explicitly weights co-accessibility scores with a distance decay function. This makes it difficult to interpret the discussion in this paragraph. I would recommend that the authors remove this section as a whole, or say that the distances observed are consistent with the assumptions of the model.

4) The approach for differential accessibility and cell type-specific accessibility is uncited and not compared to existing approaches. To identify peaks of accessibility that are differentially accessible between cell types, the authors use a Fisher's exact test. To identify peaks of accessibility that are specific to a given cell type, the authors conduct all pairwise Fisher's exact tests with the other cell types and then select sites that are significant in all tests. I'm not aware of a method like this having been implemented previously and it is unclear how well-calibrated this test is. The authors should either reference the relevant literature using this approach or should compare the performance of their method directly to existing methods for defining differential accessibility and cell type specificity of single cell ATAC-seq signals or should use an established method. The approach should be explained in more detail in the Methods as well.

5) The intersection with GWAS does not account for confounders such as LD structure. Interpreting the causal mechanisms of GWAS associations is difficult. There are many confounders including LD structure that make it hard to parse out causal variation. In fact, there's a whole body of literature on methods that try correct for these effects in overlaying GWAS signals and chromatin data (e.g. LD Score Regression or 'LDSC' - <https://doi.org/10.1038/ng.3211>). This task is further complicated by the fact that the authors are using mouse developmental data to interpret human disease signals. There are no guarantees that the patterns of development observed in the mouse will be recapitulated in humans. This is not to say that this method is without merit, but without some sort of validation, the statements that can be made in this regard are limited. There are exciting potential stories in lines 465-531, but there are also many important caveats to keep in mind. I would recommend the authors employ a more robust method (such as LDSC) and make the language more circumspect in this section (and discussion) or else consider removing the section from the manuscript. In addition, I think the length of the section is not warranted given the speculative nature. Finally, I would recommend changing the title to be more tentative about the insights into kidney disease. Again, I think using these data to glean hypotheses for further study is warranted, but none of the cross-species analysis can be considered definitive.

Minor criticism

Line 39-40: I would argue that the data here do not necessarily show a reliance of genes on distal elements (although this is already well established in the field). Instead, the authors might say something about the widespread nature of dynamics across development here.

Line 45: Saying "identified critical cell types" overstates the confidence we can have in the analysis. I would change to "implicates" or similar.

Lines 92-94: I am not sure what the authors are trying to say here. I would remove the sentence.

Lines 96-98: The authors should make it clear that single cell ATAC-seq data of the kidney has been published, but not of postnatal development.

Line 118: When the authors first describe the time points, I'd like them to provide a rationale for why these specific time points were chosen. Also, Figure 1A doesn't seem necessary to me. I would remove it.

Line 130: Do the authors mean UMAP, instead of "principal component analysis"?

Line 131: I think the authors should show what the clustering looks like without Harmony.

Line 179: I believe the general interpretation of the difference between nuclear and cellular preps is that cellular preps leave a lot of structural cell types trapped in the tissue matrix. Rather than say that cellular preps “better capture immune cells”, I would say that they are more biased towards immune cells, or something along those lines. I conceded that it’s a matter of perspective, but I think this would be a more faithful representation of the reality.

Line 183: “Finally” seems out of place here.

Lines 187-191: The reference to COVID19 does not strike me as relevant to the rest of the manuscript. I would remove this reference.

Line 201: The authors should indicate where the ChIP-seq data referenced here is coming from.

Line 202: The phrasing here is a little awkward. The authors should clarify the point.

Line 254: The authors should refer to the existing literature on single cell ATAC in the mouse kidney here too.

Line 286: “Finally” here is not necessary either.

Lines 292-293: The summary statement seems out of place to me as well.

Lines 356-357: The authors say that “this designation will need to be matched with prior cell marker-based annotations.” I’m not sure what would preclude this from having been done already. They should align these designations with existing markers and at least report a qualitative concordance.

Lines 358-360: I presume the authors are referring to GREAT here. Under default parameters, GREAT takes all the genes within a window around defined regions, so I wouldn’t describe it as using the “nearest gene”.

Lines 414-416, 462-463: The summary statements felt out of place to me.

Line 534-535: The authors should just state that this is the first single cell open chromatin map for the developing postnatal mouse kidney. I think the current sentence overstates the point.

Lines 538-539: The results shed light on potential cell types and target genes relevant to kidney disease. These all need to be validated. The authors should be clear about this here.

Lines 550-557: I believe the literature would support that this is not always the case in comparing single cell RNA and ATAC. I’d like to see the authors comment on that and maybe make a conjecture about why their data are different.

Lines 592-616: Like the GWAS results section, this section of the Discussion is too long, given the speculative nature of the underlying analysis. We know that the SNPs successfully lifted over, but we don’t know if any of the regulatory states or the roles of specific master regulators are conserved. This section should be shorter and more circumspect in language.

Line 784: The authors should define what is in the lysis buffer.

Line 809: It would be better if the authors defined the read structure for the libraries (75x75?).

Lines 826-828: I don’t think the authors mean “binarization” here. It seems like “binarization” is being used to mean counting reads in bins.

Supplementary Figures: Many of the Supplementary Figures are multiple pages long. Can they just be called different figures?

Supplementary Tables: Whether as a Supplementary Table or on the associated website, the authors should include a metadata table for RNA and ATAC. The tables should include the barcode for every cell, the UMAP coordinates for each from Fig 1, the final cell type annotation, and any relevant experimental parameters (mouse age, replicate, qc measures such as mito fraction, etc.). There should also be raw data in a convenient matrix format for others to download and analyze.

Reviewer #3 (Remarks to the Author):

The manuscript by Miao et al describes primarily the transcriptional status and changes of mouse kidney cells from day 0 to day 56 after birth using single cell RNA-seq and ATAC-seq. A main focus of the manuscript is to clarify gene expression changes seen with differentiation from nephron progenitor cells to kidney cells. The conclusions that can be drawn from the authors’ work are (i) that there are major gene expression changes following differentiation, which are associated with loss and appearance of cis-regulatory elements, and (ii) predicted transcription factors play a role in facilitating these transcriptional responses.

The strength of the study is that all of the experiments were done in primary tissues from mice, which is an interesting resource for the research community. The information obtained is very

extensive and will be very useful to the scientific community although it will be important, as the authors recognize to perform more validation of the findings in the future. Also there are many unclear extrapolations and conclusions drawn on associations that are not well defined in parts. A significant weakness of the manuscript is that it is entirely descriptive, partly predictive and complex leading to many enticing hypotheses but not a clear focus beyond the differences seen with development using only 2 time points.

The authors should do a more detailed characterization of the regulatory elements identified by sn-ATAC-seq. Moreover, the connections between the regulatory elements, predicted transcription factors and target gene expression should be developed more. Further, the last part of the analysis, human kidney disease associated SNPs to mouse kidney cell specific regulatory elements is somewhat weak albeit interesting if it turns out that mouse data can speak directly to interpretation of human SNPs. But even if it did, with human datasets only, which publicly available (e.g. Human kidney H3K27ac) why not use those?

Specific comments, suggestions and points to be addressed to further strengthen the manuscript are detailed below:

Specific points:

1. Please provide a list of marker genes for manually annotating cell types from reference 9 and 19. Are the key cell type marker genes that are shown in Figures 1d, f and S1h also the ones used for the manual annotation of the cell types? Why are the authors using different marker genes for proximal tubular cells for the sc-RNA-seq and sn-ATAC-seq data? Please add genomic distance measures in Figures 1d and S1h.
2. Please provide numbers of the correlation coefficients in Figure S1c and S1p. For P21 only one replicate is provided, why?
3. What is the percentage of overlapping peaks between bulk ATAC-seq and sn-ATAC-seq of a high abundant cell types such as proximal tubular cells or, in other words, do you find more non-overlapping regions in low abundant cell types?
4. What happens with the stromal cells, which are enriched at day 0, in the adult kidney? Do these represent mesenchymal precursors?
5. What are the number of cells and cell type percentages in sc-RNA-seq in each kidney sample – similar to Figure S1G? The Annotation in Figure S1I seems to be off. The percentage of proximal tubular cells in the sc-RNA-seq data is lower as expected ~ 30% (8.4% for PT S1, 14% for PT S3 and 8.4% for PT). Why?
6. How was the gene activity score for the top 3000 highly variable genes in sn-ATAC-seq calculated, based on TSS coverage or other peaks?
7. What are the differences (unique and overlapping) in genomic elements (distal, promoter, intron,.....) between P01, P21, P56 from Figure S2A?
8. The overlap to H3K27ac of max. 50% for each cell type seems to be low. What is the overlap compared to bulk ATAC-seq or DNase-seq for each cell type? Similar to point 1.
9. Figure 2A: Which types of elements (promoter, enhancer, ...) are unique in each cluster (cell type) for all 60,684 peaks or only the subset? Why are the authors only showing a subset of cell type-specific peaks? Is the transcription factor prediction based on the full set or only on the subset? What rationale was used to only select the shown transcription factors, p-value, literature (please provide), gene expression? An unexplained selection process might give the impression of some sort of "cherry picking". Sc-RNA-seq has a strong bias to detect mainly high abundant genes, but might miss low to medium abundant transcription factor transcripts in the cell types. Do the transcription factors bind to any of the cell type-specific peaks? It is also likely that a panel of TFs are responsible for kidney cell type-specific gene expression e.g. Hnf1b is likely a very important TF in most kidney epithelia cells, probably working together with other TFs on different cis-regulatory elements. On the other hand there is also the possibility that the selected TFs are only regulating one gene or one cis-regulatory element in a specific kidney cell. For some of the TFs ChIP-seq profiles are already available e.g. Hnf1b, Esrrg, Vdr,...
10. Among the different co-accessible elements between P0 and adult kidneys, do you also observe a significant change on gene expression levels if you assign the regulatory elements to the nearest genes? Only 74,694 (~30%) are common in P0 and adult kidneys according to your analysis.
11. page 9 line 242-250. We do not fully understand the biology of cis-regulatory elements and how they direct gene expression over large distances or even short distances. Therefore the in

silico analysis with Cicero is mainly speculative and does not add any valuable information to the manuscript.

12. Please add H3K4me3 as a marker of the promoter region to Figure S2D.

13. Do the authors use the information of cis-regulatory elements from ATAC-seq for the filtering of motif-enriched target genes derived by SCENIC or is the motif filtering based on the enrichment around the TSS?

14. Please state that you report predicted target genes for the listed Transcription Factors e.g. Fig. 2F. Are any of the predicted target genes experimentally validated? Please show snATAC-seq tracks for represented predicted target genes and the location of the transcription factor motif.

15. The enhancer element of Slc6a18 shown in Suppl. Fig. S2G could be also part of the promoter. Please show also H3K4me3 ChIP-seq mark of this region.

16. Please label ATAC-seq and ChIP-seq tracks with x kb distance additionally to the chromosome location.

17. Please add the word 'predicted' in the conclusion on page 10. 'and predicted TF-centered regulatory network'

18. The authors report that closing events are dominant during nephron progenitor differentiation. I cannot see that in Figure S2a, page 11 line 299-301; Please clarify: How many elements are getting lost from day 0 to day 21 to day 56 and how many open up from day 0 to day 21 to day 56.

19. Figure 3c and S3i: The reported target gene expression is only based on prediction. Please state this clearly in the manuscript. There is evidence that Slc34a1 is regulated by Hnf4a instead of Hnf1a. Can you find any Hnf4a motif directly underlying one of the accessible regions (ATAC-seq peaks) near Ace2? – you can add the motif information in S1Q.

20. What is the difference of the reported 15 developmental steps and the labeled stages (NP, IM, Podo, PT, LOH and DCT), Figure S3b-c?

21. Do the authors see changes in gene expression (change in fold change) of nearest genes assigned to opening chromatin (increased expression) and to closing chromatin (decreased expression) in different kidney cell types, Figure S4?

22. Would recommend exchanging Figure S4 with Figure 4. Figure 4 is primarily predictive, represents only hypotheses and should be moved into the supplemental part. Cell fate decisions and associated transcription factor needs experimental validation in future studies.

23. How many regulatory elements are unchanged between the defined differentiation stages? Do you see any changes on gene expression level of associated genes (nearest genes)?

24. What was the definition of nearest genes, distance to TSS?; How many genes do you assign to a single element?

25. Information in second paragraph on page 15 and first paragraph on page 16 is more suitable for the discussion, at least most of it.

26. Conclusion on page 16: please exchange 'found the critical role' into 'suggested a critical role'.

27. The last analysis has several major pitfalls and needs to be done in human only datasets. Through the lift-over process of human SNPs to mouse, 2/3 of the SNPs are lost. Cis-regulatory elements are less conserved compared to coding regions. The most important analysis is missing. Can you find a significant enrichment of kidney disease associated SNPs in open chromatin regions compared to random lists of SNPs? The three presented genomic regions, SHROOM3, DAB2 and UNCX are already associated with kidney function/disease, this accounts per se for no novel information. Then SNPs are not well labeled in Figure 6. Where can I find the kidney disease associated SNP? Is it the dashed line? It seems most of the SNPs are not in the open chromatin region? H3K4me3 would be also useful in Fig. 6B, D and F.

28. Please remove the CICERO-inferred co-accessibility of open chromatin regions from the ATAC-seq pictures. The interaction of regulatory elements needs to be shown experimentally with 3D chromatin conformation assays such as Hi-C, 3C, 4C

29. The last sentence includes reference to disease development. Data are derived from uninjured kidney tissue. The links to disease states are not developed in the manuscript and any reference to disease states should be removed from the manuscript.

Reviewer #4 (Remarks to the Author):

This is an outstanding paper that assesses the chromatin accessibility at a single cell/single nucleus level for developing mouse kidneys and adult kidneys. Coupled with scRNA-seq, the paper characterizes cell-type-specific DNA regulatory elements that could provide insights into kidney development and differentiation. The data allows identification of renal epithelial cell type specific enhancers and predictions of the transcription factors bound in these enhancers. The authors provide their curated data in the form of three publicly accessible web pages that will garner heavy use. These data will be extremely valuable to the research community. The following are intended to be constructive comments useful in revising the paper.

Semi-Major

1. In the cell type annotation stage, proximal tubule cells were classified into S1 and S3 cells in both scATAC-seq and scRNA-seq. Where are the S2 cells? Kap transcripts seem to be abundant in both S2 and S3 cells [PMID: 25817355 and 31689386]. In this regard, the authors' S3 could be a combination of S2 and S3. S3 cells are rare compared to S2 because they are only found in the part of the proximal tubule in the outer stripe of the outer medulla. The classic physiological function of S2 is PAH secretion [PMID: 659594]. In this regard, the PAH transporter Slc22a6 could be of value.
2. Different cell type names were interchangeably used throughout the main figures and supplementary figures. For example, PTS1 and PTS2 v.s. PT, PT_out, and PT2. This is confusing. Consistent naming of cells/cell types is desirable.
3. It seems that the authors ignored the ureteric bud-derived cells (CNT, PC, IC) in the analysis associated with Fig. 3. Are there specific reasons for this? As the author mentioned in the introduction, these cells differ from metanephric mesenchyme derived cells. How were these cells ordered in Fig. 3? Did they show a different trajectory?
4. Line 361, it is unclear how NP1-NP3 were defined. What is the basis for separating the NP into Np1-NP3? Without a clear definition in the text, it leads to confusion. The same issues were found related to Figure 4. This is an interesting figure, however, multiple new terminologies and cell types are mentioned without clear explanation and justification.
5. Line 522, the statement "the GWAS locus demonstrated a strong open chromatin region in nephron progenitors but not in any other differentiated cell types", accessible chromatin regions are seen at Uncx locus in Loop of Henle (Fig. 1D and Fig. 6E), but the weak signals could be simply "averaged" due to the fact that multiple cell types with distinct functions are present in Loop of Henle region (thin descending limb, thin ascending limb, medullary thick ascending limb, etc.). Did the author try to delineate the cell types in LOH region?
6. In viewing the scRNA-seq data at susztaklab.com/developing_adult_kidney/scRNA/, it appears that Aqp1, Aqp2, and Aqp3 have signals in most clusters suggesting that there is a lot of ambient mRNA in the samples. I think, however, that this is just a scaling problem. Would advise setting some threshold value below which cells are reported with gray color rather than dark blue.
7. While the data at susztaklab.com/developing_adult_kidney/snATAC/ are potentially very useful, searching on the basis of the bin coordinates seems impractical. It would be very nice if a bin could be selected by specifying the official gene symbol of one of the genes whose TSS is found in the bin.

... Minor

1. Line 155, "Scl12a3" should be "Slc12a3".

2. The definition of "Stroma cell" is broad. It could be resident fibroblast, vascular smooth muscle, mesangial cells, etc. Since the authors have generated the differential genes/peaks, this could be elaborated. In Fig. 1F, Were stroma 1 and 2 combined?

3. In Fig. 2a, PU.1 and Spi1 were interchangeably used.

4. Fig. S1-I, the cell annotation and values are not aligned correctly. For example, It is not clear which value was assigned to PC, etc.

5. Will the author provide the bulk adult ATAC-seq track in http://susztaklab.com/developing_adult_kidney/igv/?

6. Line 368, which "Notch" are the authors referring to?

7. Line 388, the statement "we performed immunofluorescence studies on developing kidneys (E13.5, P0, and P6)", the author provided IF for E13.5 and P6, but P0 was missing in main and supplementary figures.

8. The ligand-receptor analysis is interesting, but could be moved to supplementary figures as the main focus of the paper is chromatin accessibility.

9. recommend: 'lowly expressed' -- 'expressed at low levels'

10. The Introduction states, "The distal convoluted tubule is critical for regulated electrogenic sodium reabsorption and potassium secretion." This statement is indirectly true but likely to be misunderstood. The major route of Na reabsorption is electrically neutral since it is mediated by the Na-Cl cotransporter (NCC, Slc12a3) of the DCT and the apical plasma membrane lacks parallel ion channels. Since no potassium pathway has been identified in the apical plasma membrane of the DCT, the DCT is not believed to mediate net transepithelial K transport. It indirectly affects electrogenic transport in the collecting duct by altering delivery of Na to the principal cells.

Mark Knepper

Reviewer #1 (Remarks to the Author):

The manuscript from Professor Susztak's group contains some really extensive and important data on the epigenetic underpinnings of cell-specific developmental programmes in the murine kidney. The manuscript will be of interest to biologists, developmental biologists, nephrologists and those interested in the applications of omics technologies. The paper is beautifully written and the interpretation of the data is generally careful. However, I think that the sections pertaining to the integration of data from human genome-wide association studies with the outputs from chromatin accessibility and epigenetic marks from the mouse kidney require some clarifications.

We thank the reviewer for the positive comments. We made major changes to the section of GWAS study as suggested.

1. Zooming on three specific GWAS regions overlapping with SHROOM3, Dab2, Uncx is well justified. What I feel requires a more explicit confirmation is that the authors' analysis provides insights into mouse orthologs of human genes overlapping with GWAS CKD signals. The "ortholog" term is used only once in the legend to Figure 6.

Thank you, this is an important point that we should indeed emphasize. We added the term "ortholog" wherever needed.

2. Epigenetic marks of active transcription and gene enhancers in mouse kidney are nicely illustrated in Panel B of Figure 6. These marks should be now available for human kidneys in RoadMap Epigenomics. I wonder if this would be helpful to integrate them in Figure 6?

We thank the reviewer for this suggestion. Unfortunately, it is difficult to directly integrate human epigenetic data to the mouse locus. As per suggestions by another reviewer, we now also show the human kidney snATAC-seq data for these specific loci. We included the human kidney histone modification ChIP-seq data into this section.

3. I feel that the discussion should make a much stronger point on the need of using human kidney cells and tissues for the purpose of deciphering the biological mechanisms underlying signals of association with CKD in GWAS. The authors rightly state in Discussion the single cell transcriptome and epigenome data are necessary to understand the findings from GWAS because their molecular underpinnings are so tissue-, cell- and context specific. It is unlikely that these highly sophisticated regulatory mechanisms would be conserved between the rodent and the human. The existence of an ortholog to a potential human CKD GWAS eGene in a mouse does not necessarily mean that the identified murine epigenetic and transcriptional mechanisms can be translated to the human.

We agree with this comment. We edited the manuscript and highlighted this critical issue. We also added data from human kidney samples, and several loci showed good consistency. Future studies shall perform comprehensive analysis using human kidney tissue samples. The *Uncx* locus was only accessible in the developing kidney (NP), thus adult human data could not reveal its accessibility pattern, highlighting the importance of obtaining samples at different developmental stages.

4. Line 98 in introduction - I would not immediately dismiss the utility of “bulk tissue” in this kind of analysis given that the new bio-informatic pipelines taking advantage of the existing human cell data can adjust for potential heterogeneity in cell compositions between the samples.

We agree that recent methods such as deconvolution could help resolve the cellular heterogeneity of bulk tissue, and data obtained from “bulk tissue” do have advantages. To this end, we have deleted this sentence and adjusted the paragraph accordingly.

5. I think that the statements pertaining to eQTL analysis in human bulk tissues have to be more careful – these studies have been extremely useful in identifying kidney eGenes – the first step to understand molecular mechanisms of CKD GWAS (i.e. Nat Med. 2018;24(11):1721-1731; Nat Commun. 2018;9(1):4800.). Lines 600-601: I am not clear how single cell sequencing can overcome the linkage disequilibrium problem in the eQTL analysis.

Thank you. We have deleted the sentence pertaining eQTL. We acknowledge that single cell sequencing cannot overcome issues about LD directly, so we retained all significant SNPs within the region. We deleted the discussion about LD and focused on the findings that the overlap between GWAS SNPs and snATAC-seq peaks can help prioritize likely causal variants, cell types and genes for GWAS variants.

6. Figure 6 is very nice already but I think it would benefit from a little more specific description in the legend for all layers (i.e. SNPs, etc).

We thank the reviewer for this comment. We added more specific descriptions in the legend.

Reviewer #2 (Remarks to the Author):

In the manuscript entitled “Single cell resolution regulatory landscape of the mouse kidney highlights cellular differentiation programs and renal disease targets”, Miao and colleagues present a large resource of single cell ATAC-seq data of mouse kidneys at different ages collected on the 10X Genomics platform. Alongside single cell chromatin data, the authors also generate single cell RNA-seq data in the same time points. My expertise is in single cell genomics, not kidney biology, and so I present a critique of the methods here. This work presents a unique opportunity to understand some of the dynamics of kidney development that happen postnatally in the mouse. The authors identify relevant cell types, uncover master regulators (both known and novel), chart the likely dynamics of cell type differentiation from progenitors, and identify specific regulatory elements that are putatively important in kidney development and disease. This work is novel, and I believe it will be of general interest to the field. However, there are several issues that need to be addressed. In the main, several of the analyses should be refined or removed, and some of the relevant literature should be better referenced and compared. My major and minor criticism are presented below:

We thank the reviewer for the constructive suggestions. We adjusted our computational methods and performed additional analyses.

Major criticism

1) Single cell ATAC-seq data on the kidney already exists (either published or preprint) and is uncited in the manuscript. <https://doi.org/10.1016/j.cell.2018.06.052> is a single cell ATAC-seq atlas of the adult mouse, including kidney. <https://doi.org/10.1101/865931> is a preprint that includes mouse kidney single cell ATAC-seq data. The work by Miao et al. includes time points not previously profiled, but the authors should reference the other work and compare their results more directly. It would be interesting to see if the studies cluster together well and if the relative proportions of cell types are similar. Direct comparison of QC measures would also be informative.

We thank the reviewer for the comments. As suggested, we added the citations of prior publications and edited the paragraph.

Since the dataset in the preprint was not publicly available, we downloaded the mouse ATAC atlas data (Cusanovich et al., Cell 2018) and compared the QC metrics as suggested. In the published mouse ATAC atlas, the data have been pre-processed to keep cells with at least 500 UMIs and less than 10% mitochondria reads. (In our dataset, we kept cells with at least 1000 UMIs and less than 10% mitochondria reads. In addition, we also filtered out cells with promoter ratio below 20%). Among the kidney cells in the ATAC atlas data, the median number of peaks per cell is 4,333, which is comparable to 4,413 in our dataset.

Our dataset contains larger number of cells (28,346 in our dataset vs. 6,299 in the mouse atlas dataset), allowing us to improve the cell type resolution of the prior map. For example, we were able to identify immune and CNT cells and separate the PCT and PST segments of the

proximal tubules and IC and PC cells in the collecting duct. Thus, our dataset includes 13 annotated cell types while the previous dataset had 6. We reported the cell proportion in the mouse atlas data and our data in **Additional Table 1** and **2**, respectively. Among the annotated cells, the cell type proportion is mostly consistent.

Additional Table 1. Cell type proportions in (Cusanovich et al., Cell 2018)

cluster	11	18	25	22	1	9	16	23	30
annotation	Proximal tubule	Collecting duct, Distal Conv. Tubule, Loop of Henle	Podocyte	Glomerular Endothelial	NA*	NA*	NA*	NA*	NA*
proportion	50.15%	28.51%	7.10%	8.48%	2.65%	1.57%	0.79%	0.02%	0.73%

*NA, unannotated

Additional Table 2. Cell type proportions in our snATAC-seq dataset

cell type	PCT	PST	early PT	IC	PC	DCT	LOH	Podo	Endo	CNT	strom a2	strom a1	immu ne
proportion	35.98 %	11.21 %	1.79%	3.09%	3.62%	5.60%	14.73 %	1.48%	8.64%	4.24%	6.65%	1.57%	1.40%

To test the consistency between the two datasets, we studied the accessibility landscape around cell type-specific marker genes reported in the mouse ATAC atlas (**Additional Figures 1** and **2**). As presented below, we are able to observe good consistency between the cell type-specific peaks. (Note that CD include PC and IC cell types).

[REDACTED]

Additional Figure 1. Kidney marker genes reported in (Cusanovich et al., Cell 2018)

Additional Figure 2. Accessibility of the above marker genes in our dataset

2) A major strength of the submitted manuscript is the inclusion of time points, which allows us to observe the postnatal development of the tissue, but the study should include more time points. Capturing the correct intermediate states depends critically on capturing appropriate intermediate time points. At a minimum, the authors should justify why these three specific time points were chosen and make clear why others are not needed.

Thank you. This is an interesting point. While prior bulk studies have relied on collecting specimens at multiple time-points, single cell data can likely recover time information via the introduction of pseudo-time analysis, since cell differentiation is asynchronous and cells at multiple developmental stages are present at the same time in the developing tissue. Using computational analysis, we have recovered the developmental trajectory of different cell lineages (Figure 3).

Particularly, the P0 kidney sample contains cells starting from nephron progenitors to fully differentiated cells (Lindstrom et al., Dev Cell 2018), thus is ideal to study development. On the other hand, only fully mature, differentiated cells are expected to be present at 8 weeks of age.

Prior reports suggest that nephrogenesis in mice is completed at 3 weeks of age, therefore, we also included this time-point (McMahon et al., JASN 2008).

To specifically examine this issue, we systematically compared chromatin accessibility in each cell type collected on P0, P21 and P56 (**Additional Tables 3 and 4**). This analysis showed minimal (essentially no) differences between cells analyzed on P21 and P56. On the other hand, most cell types showed significant changes in chromatin accessibility when the P0 and P21 or P56 stages were compared.

Additional Table 3. Proportion of differentially accessible peaks between developmental time points across cell types

	p0_p21 up	p21_p0 up	p21_p56 up	p56_p21 up	p0_p56 up	p56_p0 up
PST	0.871%	1.204%	0.151%	0.977%	5.846%	6.233%
PCT	15.107%	8.276%	0.604%	1.692%	26.142%	12.826%
immune	0.000%	0.000%	0.000%	0.000%	0.000%	0.000%
DCT	0.000%	0.032%	0.000%	0.000%	0.469%	0.738%
LOH	5.789%	4.418%	0.015%	0.007%	39.134%	14.561%
stroma1	0.626%	0.637%	0.000%	0.000%	4.226%	4.034%
Endo	0.169%	0.206%	0.001%	0.000%	8.870%	4.451%
IC	0.000%	0.002%	0.002%	0.000%	0.011%	0.023%
stroma2	2.192%	1.369%	0.001%	0.001%	15.390%	10.389%
PC	0.141%	0.179%	0.001%	0.000%	5.934%	4.263%
Podo	0.000%	0.002%	0.000%	0.000%	0.244%	0.614%

Additional Table 4. Number of differentially accessible peaks between developmental time points across cell types

	p0_p21	p21_p0	p21_p56	p56_p21	p0_p56	p56_p0
PST	988	1367	171	1109	6635	7074
PCT	19102	10465	764	2139	33056	16218
immune	0	0	0	0	0	0
DCT	0	20	0	0	293	461
LOH	6987	5333	18	8	47235	17575
stroma1	648	659	0	0	4372	4174
Endo	158	193	1	0	8290	4160
IC	0	1	1	0	6	13
stroma2	3331	2080	1	1	23389	15789
PC	146	185	1	0	6129	4403
Podo	0	1	0	0	160	403

3) The analysis of Cicero link distances is inappropriate given the underlying model. In the section on lines 242-250, the authors discuss an analysis of the distance between co-accessible peaks. They make the argument that peaks up to 500kb show strong co-accessibility scores. While it's true that there are outliers with strong measures even at this distance, the general trend is of decreasing scores with increasing distance. This happens because the Cicero model explicitly weights co-accessibility scores with a distance decay function. This makes it difficult to interpret the discussion in this paragraph. I would recommend that the authors remove this section as a whole, or say that the distances observed are consistent with the assumptions of the model.

We thank the reviewer for pointing this out. We deleted this section.

4) The approach for differential accessibility and cell type-specific accessibility is uncited and not compared to existing approaches. To identify peaks of accessibility that are differentially accessible between cell types, the authors use a Fisher's exact test. To identify peaks of accessibility that are specific to a given cell type, the authors conduct all pairwise Fisher's exact tests with the other cell types and then select sites that are significant in all tests. I'm not aware of a method like this having been implemented previously and it is unclear how well-calibrated this test is. The authors should either reference the relevant literature using this approach or should compare the performance of their method directly to existing methods for defining differential accessibility and cell type specificity of single cell ATAC-seq signals or should use an established method. The approach should be explained in more detail in the Methods as well.

Thank you. Because of the binary nature of the single cell-peak matrix, Fisher's exact test has been widely used to compute differentially accessible peaks including the following high-profile papers and preprints (Chen et al., Nat Biotechnol 2019; Ziffra et al., bioRxiv 2019; Pijuan-Sala et al., Nat Cell Biol 2020; Chiou et al., bioRxiv 2019). We have now included the citations and commented in the Method section.

The pairwise comparison analysis was based on our previous publication (Park et al., Science 2018), and another published snATAC-seq paper (Pijuan-Sala et al., Nat Cell Biol 2020). We found that the pairwise comparison improves the stringency of cell type-specific gene or peak identification, which was useful for our downstream analysis.

5) The intersection with GWAS does not account for confounders such as LD structure. Interpreting the causal mechanisms of GWAS associations is difficult. There are many confounders including LD structure that make it hard to parse out causal variation. In fact, there's a whole body of literature on methods that try correct for these effects in overlaying GWAS signals and chromatin data (e.g. LD Score Regression or 'LDSC' - <https://doi.org/10.1038/ng.3211>). This task is further complicated by the fact that the authors are using mouse developmental data to interpret human disease signals. There are no guarantees that the patterns of development observed in the mouse will be recapitulated in humans. This is not to say that this method is without merit, but without some sort of validation, the statements that can be made in this regard are limited. There are exciting potential stories in lines 465-531, but there are also many important caveats to keep in mind. I would recommend the authors employ a more robust method (such as LDSC) and make the

language more circumspect in this section (and discussion) or else consider removing the section from the manuscript. In addition, I think the length of the section is not warranted given the speculative nature. Finally, I would recommend changing the title to be more tentative about the insights into kidney disease. Again, I think using these data to glean hypotheses for further study is warranted, but none of the cross-species analysis can be considered definitive. Thank you for making this important point.

We agree that the LD structure presents an important limitation for causal variant prioritization. We believe that it is not appropriate to run LDSC using the mouse lift-over dataset, due to multiple uncertainties of orthologous mapping.

As a compromise, we now also included human kidney snATAC-seq data. Our results indicate reasonable conservation of chromatin accessibility between the human and mouse locus. Future studies shall formally analyze this issue.

We believe that the selected examples of the mouse kidney snATAC-seq data can be a reference for human GWAS prioritization. In addition, the inclusion of different developmental stages is important, as illustrated by the *Uncx* locus.

Minor criticism

Line 39-40: I would argue that the data here do not necessarily show a reliance of genes on distal elements (although this is already well established in the field). Instead, the authors might say something about the widespread nature of dynamics across development here.

Thank you for the comments. In our analysis, several findings indicated the reliance of distal elements. First, we showed distal regulatory elements and intronic regions account for a large proportion of open chromatin in both single cell data and bulk data. Second, we showed that up to 50% of open chromatin regions are covered by either P0 or adult H3K27ac ChIP-seq signals, indicating a large number of active enhancer regions. Third, we showed that for some genes including *Umod*, the specificity of gene expression correlated with an enhancer peak rather than its TSS accessibility.

Line 45: Saying “identified critical cell types” overstates the confidence we can have in the analysis. I would change to “implicates” or similar.

We made this edit.

Lines 92-94: I am not sure what the authors are trying to say here. I would remove the sentence.

We made this edit.

Lines 96-98: The authors should make it clear that single cell ATAC-seq data of the kidney has been published, but not of postnatal development.

We made this edit.

Line 118: When the authors first describe the time points, I'd like them to provide a rationale for why these specific time points were chosen. Also, Figure 1A doesn't seem necessary to me. I would remove it.

We made this edit. We adjusted Figure 1A to be more informative.

Line 130: Do the authors mean UMAP, instead of "principal component analysis"?

It is PCA. PCA was used to reduce dimensions and UMAP was used for visualizing PCA space. We have deleted this sentence but included additional details in the Methods section.

Line 131: I think the authors should show what the clustering looks like without Harmony.

Additional Figures 5 and 6 below show the scRNA and snATAC data without Harmony integration. We also included these figures in the Supplemental Figures.

Additional Figure 3. snATAC-seq clustering without and with Harmony

Additional Figure 4. scRNA-seq clustering without and with Harmony

Line 179: I believe the general interpretation of the difference between nuclear and cellular preps is that cellular preps leave a lot of structural cell types trapped in the tissue matrix.

Rather than say that cellular preps “better capture immune cells”, I would say that they are more biased towards immune cells, or something along those lines. I conceded that it’s a matter of perspective, but I think this would be a more faithful representation of the reality.
Thank you for the suggestion. We agree and made this edit.

Line 183: “Finally” seems out of place here.
We made this edit.

Lines 187-191: The reference to COVID19 does not strike me as relevant to the rest of the manuscript. I would remove this reference.
We made this edit.

Line 201: The authors should indicate where the ChIP-seq data referenced here is coming from.
We included the reference.

Line 202: The phrasing here is a little awkward. The authors should clarify the point.
We made this edit.

Line 254: The authors should refer to the existing literature on single cell ATAC in the mouse kidney here too.
Thank you. In the existing dataset, cell type-specific TFs have not been studied.

Line 286: “Finally” here is not necessary either.
We made this edit.

Lines 292-293: The summary statement seems out of place to me as well.
We made this edit.

Lines 356-357: The authors say that “this designation will need to be matched with prior cell marker-based annotations.” I’m not sure what would preclude this from having been done already. They should align these designations with existing markers and at least report a qualitative concordance.
Thank you for pointing this out. We included additional analysis and was able to match our designation with prior annotations. We added a heatmap in the **Figure 4b** and discussed in the main text.

Specifically, NP3 is matched to renal vesicles based on the enrichment of *Wnt4* (Park et al., Dev. 2007), IM1-2 and Podo1 match Comma- and S- shaped body signatures, such as *Notch1*, *Notch2*, *Jag1*, and *Lfng* (Lindstrom et al., J Am Soc Nephrol 2018). Thus IM1-2 and Podo1 may represent CSB/SSB that committed to tubule or podocyte lineages, respectively. However, these nephron progenitor subgroups (or states) need to be functionally validated.

Lines 358-360: I presume the authors are referring to GREAT here. Under default parameters, GREAT takes all the genes within a window around defined regions, so I wouldn't describe it as using the "nearest gene".

Thank you, we made this edit.

Lines 414-416, 462-463: The summary statements felt out of place to me.

We made this edit.

Line 534-535: The authors should just state that this is the first single cell open chromatin map for the developing postnatal mouse kidney. I think the current sentence overstates the point.

We changed the sentence accordingly.

Lines 538-539: The results shed light on potential cell types and target genes relevant to kidney disease. These all need to be validated. The authors should be clear about this here.

Thank you, we adjusted this statement.

Lines 550-557: I believe the literature would support that this is not always the case in comparing single cell RNA and ATAC. I'd like to see the authors comment on that and maybe make a conjecture about why their data are different.

Thank you for pointing this out. We agree that this statement requires further systematic analysis. Geometrical relationships within a low-dimensional embedding must be interpreted with caution, but we found more distinct clusters with snATAC-seq data compared with our scRNA-seq data. Prior published data that analyzed the developing mouse and human kidney indicated very significant relatedness between the developing kidney cells. We reasoned that this might be due to the continuous nature of cellular development reflected by the continuous changes in gene expression (**Additional Figure 5**). On the other hand, snATAC-Seq data obtained at the same time highlighted very distinct cell clusters. As discussed in the text, this could be attributed to the fact that open chromatin data could better reflect the regulatory landscape and is more discrete in nature.

In the brain, Lake et al., (Nat Biotechnol 2018) observed higher cellular resolution in the snDrop-seq data (that measures RNA) than scTHS-seq (that measures open chromatin). Their different observations could be due to 1. Differences in the biological samples (developing vs post mitotic); 2. Cell cycle genes can confound single cell RNA-seq data, while the open chromatin stages are more stable, as suggested by (Ma et al., BioRxiv 2020); 3. Differences in technology between our data and publication by Lake et al (different noise as well as sequence depth).

We have adjusted the text accordingly.

[REDACTED]

Additional Figure 5. Clustering results of scRNA-seq data of developing mouse and human kidneys (left: Combes et al., Development 2019; right: Lindstrom et al., Dev Cell 2018)

Lines 592-616: Like the GWAS results section, this section of the Discussion is too long, given the speculative nature of the underlying analysis. We know that the SNPs successfully lifted over, but we don't know if any of the regulatory states or the roles of specific master regulators are conserved. This section should be shorter and more circumspect in language.

Thank you, we shortened the paragraph and adjusted our language.

Line 784: The authors should define what is in the lysis buffer.

We added this information in the methods. (The lysis buffer contains 10 mM Tris HCl pH 7.4, 10 mM NaCl, 3 mM MgCl₂, and 0.1% Nonidet™ P40 Substitute in nuclease-free water.)

Line 809: It would be better if the authors defined the read structure for the libraries (75x75?). The read structure of snATAC-seq is 50x50, the read structure of bulk ATAC-seq is 150x150 and that of scRNA-seq is 150x150. We include this information in the method section.

Lines 826-828: I don't think the authors mean "binarization" here. It seems like "binarization" is being used to mean counting reads in bins.

Thanks for pointing this out. We corrected the description: "after binning the fragments into 5 kb windows."

Supplementary Figures: Many of the Supplementary Figures are multiple pages long. Can they just be called different figures?

Thanks for pointing this out, we reorganized the supplemental figures.

Supplementary Tables: Whether as a Supplementary Table or on the associated website, the authors should include a metadata table for RNA and ATAC. The tables should include the barcode for every cell, the UMAP coordinates for each from Fig 1, the final cell type annotation,

and any relevant experimental parameters (mouse age, replicate, qc measures such as mito fraction, etc.). There should also be raw data in a convenient matrix format for others to download and analyze.

Thank you. Metadata tables including barcodes, UMAP coordinates, cell type annotation, developmental stage, and QC measures for both scRNA-seq and snATAC-seq, as well as bulk ATAC-seq datasets have been uploaded together with the raw data to GEO with accession number GSE157079. The processed data are also available in our interactive websites.

Reviewer #3 (Remarks to the Author):

The manuscript by Miao et al describes primarily the transcriptional status and changes of mouse kidney cells from day 0 to day 56 after birth using single cell RNA-seq and ATAC-seq. A main focus of the manuscript is to clarify gene expression changes seen with differentiation from nephron progenitor cells to kidney cells. The conclusions that can be drawn from the authors' work are (i) that there are major gene expression changes following differentiation, which are associated with loss and appearance of cis-regulatory elements, and (ii) predicted transcription factors play a role in facilitating these transcriptional responses.

The strength of the study is that all of the experiments were done in primary tissues from mice, which is an interesting resource for the research community. The information obtained is very extensive and will be very useful to the scientific community although it will be important, as the authors recognize to perform more validation of the findings in the future. Also there are many unclear extrapolations and conclusions drawn on associations that are not well defined in parts. A significant weakness of the manuscript is that it is entirely descriptive, partly predictive and complex leading to many enticing hypotheses but not a clear focus beyond the differences seen with development using only 2 time points.

The authors should do a more detailed characterization of the regulatory elements identified by sn-ATAC-seq. Moreover, the connections between the regulatory elements, predicted transcription factors and target gene expression should be developed more. Further, the last part of the analysis, human kidney disease associated SNPs to mouse kidney cell specific regulatory elements is somewhat weak albeit interesting if it turns out that mouse data can speak directly to interpretation of human SNPs. But even if it did, with human datasets only, which publicly available (e.g. Human kidney H3K27ac) why not use those?

We thank the reviewer for the overall comments of the manuscript. We do hope our manuscript will be a useful resource for the community to understand kidney disease and aid studies related to kidney development and epigenomics. We agree that the primary focus of this manuscript lies in data generation and integrative analysis.

We have taken efforts to highlight critical new biological insights. For example, we performed gene regulatory inference analysis and cell type specific TF enrichment analysis. We also highlighted nephron progenitor differentiation and key transcription factors that show differences during development. We validated the role of *Foxl1* in podocytes, which has not been identified by earlier studies. We highlighted the importance of using snATAC-seq data to prioritize causal variants cell types and genes for human GWAS variants. In the revised version, we also show cell type and developmental stage specific differential accessibility peaks. Finally, we have made the information publicly available not only via GEO but also via an interactive website.

Specific comments, suggestions and points to be addressed to further strengthen the manuscript are detailed below:

Specific points:

1. Please provide a list of marker genes for manually annotating cell types from reference 9 and 19. Are the key cell type marker genes that are shown in Figures 1d, f and S1h also the ones used for the manual annotation of the cell types? Why are the authors using different marker genes for proximal tubular cells for the sc-RNA-seq and sn-ATAC-seq data? Please add genomic distance measures in Figures 1d and S1h.

We thank the reviewer for these comments.

1. We now provide a table with marker genes from References 9 and 19 that was used for manual annotation of clusters (**Supplemental Table 2**). Genes shown in Figures 1d, f and S1h are from this list and these genes were used for cell type annotation.

2. It is expected that ATAC-seq and RNA-seq generates somewhat complementary information, such as the higher depth of the epigenome data and its likely better cell type specificity, however, open chromatin information does not easily translate into RNA expression (given the role of distal regulatory elements). The example figure showed marker genes where gene expression could be inferred based on open chromatin around the TSS, while others were more complex such as *Umod*. The full list of DEGs and DAPs are listed in **Tables S2&5**. Based on the selected markers and downstream analysis, we believe our annotations are reliable. For more marker information, our interactive website could be useful.

3. We now included genomic distance measures in Figures 1d and S4a (former Figure S1h).

2. Please provide numbers of the correlation coefficients in Figure S1c and S1p. For P21 only one replicate is provided, why?

Thanks for the comments. We show the correlation coefficients in the **Additional Tables 5 and 6**.

In mice, kidney development completes around day 21, therefore we observed minimal differences between P56 and P21 (**Additional Tables 3-4**), hence only one sample from P21 was sequenced.

Additional Table 5. Correlation coefficients in Figure S1c

	P0.1	P0.2	P21.1	P21.2	P56.1	P56.2
P56_batch_1	0.7368	0.8068	0.8377	0.8512	0.8599	0.8554
P56_batch_2	0.7276	0.7942	0.8308	0.8404	0.8505	0.8482
P21_batch_1	0.744	0.8257	0.8444	0.8496	0.8251	0.831
P0_batch_1	0.8586	0.9097	0.8657	0.8781	0.836	0.8416
P0_batch_2	0.8567	0.9156	0.868	0.8805	0.8353	0.8422

Additional Table 6. Correlation coefficients in Figure S1p

	NP	Podo	PCT	PST	LOH	DCT	PC	IC	stroma	Endo	immune
NP	0.426	0.132	-0.186	-0.142	0.069	-0.123	-0.054	-0.154	0.147	0.043	-0.022

Podo	0.183	0.534	-0.164	-0.133	-0.020	-0.130	-0.078	-0.135	0.075	0.078	-0.079
PT S1	-0.067	-0.023	0.502	0.344	-0.048	-0.124	-0.121	-0.114	-0.106	-0.171	-0.141
PT S3	-0.231	-0.168	0.727	0.641	-0.156	-0.113	-0.191	-0.104	-0.253	-0.229	-0.140
LOH	-0.154	-0.091	0.028	0.057	0.328	0.391	0.108	0.177	-0.237	-0.226	-0.114
DCT	-0.149	-0.121	-0.033	-0.015	0.222	0.512	0.102	0.303	-0.229	-0.189	-0.145
PC	-0.084	-0.133	-0.130	-0.093	0.088	0.166	0.495	0.250	-0.109	-0.112	-0.109
IC	-0.141	-0.088	-0.069	-0.065	0.006	0.215	0.156	0.524	-0.161	-0.132	-0.109
stroma	0.258	0.092	-0.259	-0.221	-0.052	-0.208	-0.059	-0.216	0.595	0.164	-0.055
Endo	0.005	0.002	-0.268	-0.237	-0.121	-0.184	-0.082	-0.157	0.157	0.606	0.101
immune	-0.088	-0.093	-0.220	-0.205	-0.148	-0.125	-0.088	-0.140	-0.044	0.063	0.635

3. What is the percentage of overlapping peaks between bulk ATAC-seq and sn-ATAC-seq of a high abundant cell types such as proximal tubular cells or, in other words, do you find more non-overlapping regions in low abundant cell types?

The number and percentage of overlapping peaks between snATAC-seq and bulk ATAC-seq are provided in **Additional Tables 7 and 8**.

In general, we observed greater overlap between bulk ATAC and common cell types, such as PT and DCT, than rare cell types, such as immune and stromal cells in the adult kidney.

Due to differences in cell fraction between the developing and adult kidneys, we observed a greater overlap with NP cells and podocytes in P0 kidneys.

Additional Table 7. Number of overlapped peaks between each cell type and bulk peaks

	PCT	PST	DCT	PC	LOH	stroma2	stroma1	IC	NP	Endo	immune	Podo
total_N_peaks	117853	106309	58352	96012	111406	138878	96002	53247	127933	87286	33371	61124
P56-2	96599	87332	51222	65427	80094	65276	54670	43823	68026	53864	22404	42982
P56-1	101301	90180	52819	68743	84686	69477	57700	44887	72308	57684	23201	44485
P21-2	88197	79395	51854	69892	84005	68879	57733	44987	72362	53889	22118	45304
P21-1	81821	74869	49986	64555	78076	63948	54186	43024	67066	51299	21513	42995
P0-2	79620	70851	47088	70707	84764	89745	71859	41665	93442	61157	23471	51290
P0-1	75085	67505	47696	73269	85395	91402	73743	42262	91316	58936	22382	50416

Additional Table 8. Proportion of overlapped peaks between each cell type and bulk peaks

	PCT	PT2	DCT	PC	LOH	stroma2	stroma1	IC	NP	Endo	immune	Podo
P56-2	0.82	0.82	0.88	0.68	0.72	0.47	0.57	0.82	0.53	0.62	0.67	0.70
P56-1	0.86	0.85	0.91	0.72	0.76	0.50	0.60	0.84	0.57	0.66	0.70	0.73
P21-2	0.75	0.75	0.89	0.73	0.75	0.50	0.60	0.84	0.57	0.62	0.66	0.74
P21-1	0.69	0.70	0.86	0.67	0.70	0.46	0.56	0.81	0.52	0.59	0.64	0.70
P0-2	0.68	0.67	0.81	0.74	0.76	0.65	0.75	0.78	0.73	0.70	0.70	0.84
P0-1	0.64	0.63	0.82	0.76	0.77	0.66	0.77	0.79	0.71	0.68	0.67	0.82

4. What happens with the stromal cells, which are enriched at day 0, in the adult kidney? Do these represent mesenchymal precursors?

We thank the reviewer for this interesting question. Indeed, the term “stromal cell” is rather broad and serves as substitute term for a multitude of different cell types. We conducted additional analyses for cells contained in the stroma clusters in the scRNA-seq data and in the snATAC-seq data.

In the scRNA-seq data, we found heterogenous groups (**Figures S8-9**), which is consistent with recent findings that renal interstitium of the developing kidney shows remarkable cellular heterogeneity (England et al., Development 2020, doi:10.1242/dev.190108). Some cells demonstrated immune cell-associated markers (*C1qb*, *Cx3cr1*), others showed markers of fibroblasts (e.g. *Col3a1*), smooth muscle cells (e.g. *Acta2*), mesangial cells (e.g. *Gata3*), and a few cells with a juxtaglomerular signature (e.g. *Akr1b7*) or tubule-associated markers (e.g. *Lrp2*, *Umod*, *Slc12a3*, *Aqp2*). It is interesting to note that such a various mixture of cells clustered very distinctly together in the UMAP projections when analyzing all cells together. However, as subclustering revealed overlapping instead of distinct signatures, we preferred the broader term “stroma cells”, concordant with previous single cell publications.

We also examined the P0 stroma population in the snATAC-seq data. We conducted subclustering analysis using latent semantic indexing-based methods with Seurat and obtained 16 subclusters (**Additional Figure 6**). However, by examining the marker genes from our scRNA-seq subclustering results and a reference paper (Combes et al., Development 2019), we did not observe major differences in gene activity scores (**Additional Figure 8**).

Subclustering of the snATAC data is shown in **Additional Figures 6-8**. The heatmap of differentially accessible peaks is shown in **Additional Figure 9**.

Additional Figure 6. UMAP embeddings of stroma subclustering results

Additional Figure 7. Dot plot of gene activity scores of marker genes in P0 snATAC-seq data stroma subclusters

[REDACTED]

Additional Figure 8. Comparison of stroma subcluster specific genes between (A) gene expression in annotated scRNA-seq data from (Combes et al., Development 2019) and (B) gene activity scores in snATAC-seq data

As a result of this heterogeneous nature of the various stroma cell subtypes, in our view, subclustering of stroma cells did not yield signatures robust enough for us to feel comfortable about the specific cell identities. Of note, P0 and adult stroma showed large differences, which might be related to the underlying biology of batch effect. We, therefore, are unable to answer the question of the reviewer with complete fidelity.

We refer the interested reader to the **Figures S8-9** and **Supplemental Table 4** for comparison of P0 and adult stroma cell signatures and DEGs. More targeted studies will be needed to further delineate this differentiation path, although we feel that such analyses are beyond the scope and focus of the current manuscript.

5. What are the number of cells and cell type percentages in sc-RNA-seq in each kidney sample – similar to Figure S1G? The Annotation in Figure S1I seems to be off.

The scRNA-seq dataset incorporated 1 adult and 1 P0 sample. The bar graphs in **Figure S4a** (former Figure S1i) now visualize individual sample fractions. We adjusted the annotation legend accordingly.

The percentage of proximal tubular cells in the sc-RNA-seq data is lower as expected ~ 30% (8.4% for PT S1, 14% for PT S3 and 8.4% for PT). Why?

Thank you for noticing. This is actually biologically important. The lower overall percentage for PT was due to the lower percentage of PT cells at the P0 stage. PT cells proliferate and undergo maturation after birth contributing to the growth of the kidney.

We have split the numbers and percentages displayed in the graph. For the adult sample, the percentage of all PT cells is 60.9%, which is in concordance with what we see from our previous analyses (Park et al., Science 2018, 60.5% of PT cells). In P0, both absolute numbers and percentages of all PT cells combined are lower (~15% PT cells).

6. How was the gene activity score for the top 3000 highly variable genes in sn-ATAC-seq calculated, based on TSS coverage or other peaks?

It is described in the methods section. Briefly, it is calculated by aggregating reads in the gene body and 2kb upstream. This method was also the default method implemented in Seurat, a widely used single cell analysis package.

7. What are the differences (unique and overlapping) in genomic elements (distal, promoter, intron,.....) between P01, P21, P56 from Figure S2A?

Thanks for this comment. We studied the shared and unique genomic elements across different bulk samples from **Figure S11a** (former Figure S2a) (**Additional Figure 9**). As expected, most of the peaks are accessible across different stages. Interestingly, there are more P0 specific open chromatin in distal and intronic elements. By contrast, the exonic regions and 5'-UTR show more conservative patterns across developmental stages.

Additional Figure 9. Shared and unique genomic elements across bulk samples

8. The overlap to H3K27ac of max. 50% for each cell type seems to be low. What is the overlap compared to bulk ATAC-seq or DNase-seq for each cell type? Similar to point 1.

Since there are accessible regions beyond active enhancers, we think the 50% overlap of H3K27ac and ATAC-seq peaks is reasonable and consistent with prior results. Other open chromatin regions might be related to baseline gene expression machinery. By comparing with bulk data, we found 35.18% of P0 bulk ATAC-seq peaks (38,690 out of 109,987) overlap with P0 H3K27ac peaks, and 36.73% of P56 bulk ATAC-seq peaks (46,626 out of 126,951) overlap with P56 H3K27Ac peaks. This is also in concordance with our snATAC-seq data.

9. Figure 2A: Which types of elements (promoter, enhancer, ...) are unique in each cluster (cell type) for all 60,684 peaks or only the subset? Why are the authors only showing a subset of cell type-specific peaks? Is the transcription factor prediction based on the full set or only on the subset? What rationale was used to only select the shown transcription factors, p-value, literature (please provide), gene expression? An unexplained selection process might give the impression of some sort of “cherry picking”. Sc-RNA-seq has a strong bias to detect mainly high abundant genes, but might miss low to medium abundant transcription factor transcripts in the cell types. Do the transcription factors bind to any of the cell type-specific peaks? It is also likely that a panel of TFs are responsible for kidney cell type-specific gene expression e.g. Hnf1b is likely a very important TF in most kidney epithelia cells, probably working together with other TFs on different cis-regulatory elements. On the other hand there is also the possibility that the selected TFs are only regulating one gene or one cis-regulatory element in a specific kidney cell. For some of the TFs ChIP-seq profiles are already available e.g. Hnf1b, Esrrg, Vdr,...

Thank you.

1. The distribution of different regulatory elements (promoter, enhancer, etc.) among the cell type specific peaks (60,684 in total) is shown in **Figure S11**.

2. In the original submission, we showed the top 2,000 cell type-specific peaks for each cell type for aesthetics. We now replaced the figure to contain all significant cell type specific DARs (**Figure 2a**). The full list of cell type-specific peaks can be found in **Supplemental Table 5**.

3. All further downstream analyses were conducted on these full 60,684 peaks without any additional filtering.

4. The transcription factor enrichment analysis was conducted by HOMER with the cell type-specific peaks. HOMER analysis is based on motif enrichment. However, some transcription factors bind to similar motifs that HOMER cannot discriminate. Thus, we used gene expression data to support the enrichment of a cell-type specific transcription factor (**Figure 2c**).

5. In addition, we have conducted SCENIC analysis to identify cell type specific transcription factors and target genes using the scRNA-seq data.

Our integrative analysis provided complementary results. While transcription factors might be expressed at low levels, therefore missed by RNA analysis, HOMER is unable to computationally distinguish between closely related transcription factor motifs. Integrating different modalities helped us identify key gene regulation networks more faithfully.

10. Among the different co-accessible elements between P0 and adult kidneys, do you also observe a significant change on gene expression levels if you assign the regulatory elements to the nearest genes? Only 74,694 (~30%) are common in P0 and adult kidneys according to your analysis.

Thanks for pointing this out. We have removed this section as suggested by another reviewer. We acknowledge that the analysis here has several limitations, such as the heuristic cut-off used by Cicero and differences in cell type distribution difference between P0 and adult as well as potential batch effect.

11. page 9 line 242-250. We do not fully understand the biology of cis-regulatory elements and how they direct gene expression over large distances or even short distances. Therefore, the in silico analysis with Cicero is mainly speculative and does not add any valuable information to the manuscript.

Thank you, we adjusted this statement.

12. Please add H3K4me3 as a marker of the promoter region to Figure S2D.

We added H3K4me3 tracks in the corresponding section.

13. Do the authors use the information of cis-regulatory elements from ATAC-seq for the filtering of motif-enriched target genes derived by SCENIC or is the motif filtering based on the enrichment around the TSS?

Thanks for the comment. The motif filtering step of our SCENIC analysis was based on the enrichment around the TSS. SCENIC filtered motifs with two gene-motif rankings: 10 kb around the TSS and 500 bp upstream. The ATAC-seq information was not used for the SCENIC analysis.

14. Please state that you report predicted target genes for the listed Transcription Factors e.g. Fig. 2F. Are any of the predicted target genes experimentally validated? Please show snATAC-seq tracks for represented predicted target genes and the location of the transcription factor motif.

Thank you for this comment. We clarified this point in the figures and the manuscript text. To our knowledge, these genes are not experimentally validated. We included snATAC-seq tracks for the predicted target genes in **Fig. S15**. From the tracks, we can see that the predicted target genes are also accessible in NP cluster. With regards to the location of the transcription factor motif, we found that there are usually multiple potential TF binding sites around the gene, and we are not able to nominate one specific location (**Additional Figure 10**). To this end, we acknowledge the limitation of SCENIC analysis in the main text.

Additional Figure 10. Chromatin accessible landscape around *Eya1* locus and *Uncx* motif binding sites

15. The enhancer element of *Slc6a18* shown in Suppl. Fig. S2G could be also part of the promoter. Please show also H3K4me3 ChIP-seq mark of this region.

Thanks for pointing this out. We added P0 and P56 H3K4me3 and H3K427ac tracks to **Figure S12c** (former Figure S2g). The distal element of *Slc6a18* overlaps with P56 H3K427ac signal but not H3K4me3 signal, suggesting that this is enhancer region. In addition, this region was reported as a distal regulatory region in a prior publication (Cao et al., Science 2018).

16. Please label ATAC-seq and ChIP-seq tracks with x kb distance additionally to the chromosome location.

Thanks, we have labeled ATAC-seq and ChIP-seq tracks with the respective distance measures throughout all Figures and Supplemental Figures.

17. Please add the word 'predicted' in the conclusion on page 10. 'and predicted TF-centered regulatory network'

Thank you. We took out the summary statement, as another reviewer felt it was redundant.

18. The authors report that closing events are dominant during nephron progenitor differentiation. I cannot see that in Figure S2a, page 11 line 299-301; Please clarify: How many elements are getting lost from day 0 to day 21 to day 56 and how many open up from day 0 to day 21 to day 56.

Thanks for pointing out the typo. The reference should be **Figure S11d** (former **Figure S2c**). We can see that NP has more peaks accessible compared with LOH and podocytes. Although the number of accessible peaks in PT is comparable with that in NP, we reason that this was likely due to the high abundance of PT as well as the heterogeneity of PT (including PCT and PST).

The number of peaks gaining or losing accessibility during differentiation is shown in **Figure S18** (former **Figure S4**). The bar charts alongside each arrow indicated the number of peaks differentially accessible between two stages. We adjusted the figure legends to make it clearer. In addition, the pairwise comparison of accessible peaks was provided in **Additional Tables 3-4**.

19. Figure 3c and S3i: The reported target gene expression is only based on prediction. Please state this clearly in the manuscript. There is evidence that *Slc34a1* is regulated by *Hnf4a* instead of *Hnf1a*. Can you find any *Hnf4a* motif directly underlying one of the accessible regions (ATAC-seq peaks) near *Ace2*? – you can add the motif information in S1Q.

Thank you. We adjusted figure legends and main text to make it clear that target gene expression is based on prediction.

We agree with the reviewer that *Slc34a1* is a predicted target gene of not only *Hnf1a*, but also other TFs including *Hnf4a*, as inferred in our analysis using SCENIC (**Additional Table 9**):

Additional Table 9. SCENIC predicted TFs that target *Slc34a1*

TF	gene	nMotifs	bestMotif	NES*
Hnf1a	Slc34a1	101	hocomoco__HNF1A_HUMAN.H11MO.0.C	9.23
Hnf1b	Slc34a1	23	hocomoco__HNF1B_HUMAN.H11MO.1.A	5.58
Hnf4a	Slc34a1	218	transfac_pro__M03828	7.27
Maf	Slc34a1	1	dbcorrd__NFE2__ENCSR000FAF_1__m1	3.12

*NES, normalized enrichment scores

This information is included in **Supplemental Table 7** (Regulon target information). We chose to depict *Slc34a1* as a target of *Hnf1a* because the normalized enrichment score (NES) was highest for this TF.

Also, we thank the reviewer for highlighting that *Ace2* is a predicted target gene of *Hnf4a*. However, the motif binding site is located within another gene, so additional validation is needed to support the regulatory relationship.

20. What is the difference of the reported 15 developmental steps and the labeled stages (NP, IM, Podo, PT, LOH and DCT), Figure S3b-c?

The cell differentiation data in **Figures S16b-c** (former **Figures S3b-c**) was generated using the unbiased cell trajectory analysis, such as Monocle and RNA velocity. Based on the inferred pseudotime of snATAC-seq data, we further divided each group into subgroups, labelled as NP1-NP3 and so on. Thus, the 15 developmental stages and labeled stages in former Figures S3b-c were from the same pseudotime results. We performed differential accessibility analysis and gene ontology analysis for these higher resolution developmental stages and this information is presented in **Figure 4** and **Figure S18** (former **Figure S4**).

21. Do the authors see changes in gene expression (change in fold change) of nearest genes assigned to opening chromatin (increased expression) and to closing chromatin (decreased expression) in different kidney cell types, Figure S4?

Thank you, some of the gene expression changes along different cell type trajectories are shown in **Figures 3, S17b** (former S3i) and **S21** (former S4c). In the text, we reported that the chromatin changes do not always correlate with gene expression and showed the example of *Wt1*.

Here, we showed additional examples of gene expression changes along the 3 trajectories in **Additional Figure 11**. Taken together, we found most of the gene expressions follow the same trend as open chromatin.

Additional Figure 11. Examples of gene expression dynamics along trajectory

22. Would recommend exchanging Figure S4 with Figure 4. Figure 4 is primarily predictive, represents only hypotheses and should be moved into the supplemental part. Cell fate decisions and associated transcription factor needs experimental validation in future studies. We appreciate this suggestion, however, **Figure S18** (former **Figure S4**) is based on the same data trajectory and contains very similar information to **Figure 4**. **Figure S18** is a further predictive model that uses GREAT nominated target genes from the differential accessible peaks, while **Figure 4** shows the dynamic open chromatin changes and highlights the transcription factors. We have validated a selected list of transcription factors such as Foxl1, HNF4A and TFAP2B. We would therefore argue that **Figure 4** represents a key information of this manuscript.

23. How many regulatory elements are unchanged between the defined differentiation stages? Do you see any changes on gene expression level of associated genes (nearest genes)? Thank you for the interesting questions. The first question is a biologically important but complicated question. It is difficult to derive a statistical test under the null hypothesis that “the group mean values are different”. In addition, the relatively small number of cells in each stage imposed additional uncertainty to the analysis. To this end, we studied the conserved and distinct peaks in a larger scale, the cell type scale. In **Figure S11d**, we showed that more than 30,000 peaks were shared between NP, PT, LOH, and DCT clusters.

For the second question, we did see changes in gene expression of associated genes, and some examples were shown in **Figures 3 and S17b**. We present additional examples in **Additional Figure 10**. As discussed in the text, there are consistent and asynchronized patterns between gene expression and chromatin accessibility. It is worth noting that due to the limitations of

snATAC-seq and scRNA-seq data such as the uneven drop-out, it is difficult to definitively answer this question.

24. What was the definition of nearest genes, distance to TSS?; How many genes do you assign to a single element?

Thank you for pointing this out. We changed the text to nearby genes as it is a more accurate description of GREAT analysis. The definition of nearby genes was based on distance to TSS. Genes from -5kb to +1kb were assigned to the element.

25. Information in second paragraph on page 15 and first paragraph on page 16 is more suitable for the discussion, at least most of it.

We made these changes.

26. Conclusion on page 16: please exchange 'found the critical role' into 'suggested a critical role'.

Thank you. We took out the summary statement, as another reviewer felt it was redundant.

27. The last analysis has several major pitfalls and needs to be done in human only datasets. Through the lift-over process of human SNPs to mouse, 2/3 of the SNPs are lost. Cis-regulatory elements are less conserved compared to coding regions. The most important analysis is missing. Can you find a significant enrichment of kidney disease associated SNPs in open chromatin regions compared to random lists of SNPs? The three presented genomic regions, SHROOM3, DAB2 and UNCX are already associated with kidney function/disease, this accounts per se for no novel information. Then SNPs are not well labeled in Figure 6. Where can I find the kidney disease associated SNP? Is it the dashed line? It seems most of the SNPs are not in the open chromatin region? H3K4me3 would be also useful in Fig. 6B, D and F.

Thank you for making this important point. While cis-regulatory elements are less conserved than coding elements, they are more conserved than other genomic regions. Indeed, conservation is often used to nominate cis-regulatory elements.

We apologize for the misunderstanding, we only plotted significant GWAS SNP, therefore all plotted SNPs passed significance threshold. Indeed, this is a critical point that using snATAC-seq data, most significant GWAS SNPs are not in open chromatin region, those within open chromatin regions are more likely to be regulatory and functional. Therefore, the snATAC-seq data can provide critical information to nominate potential cell types, causal variants and target genes.

We agree that human data is critical, therefore, we now also included human kidney snATAC-seq data (**Figure 6**). Our results indicate remarkable conservation between the human and mouse locus. Future studies shall perform formal analysis on the conservation. We believe that in the select examples, the mouse kidney snATAC-seq data can be informative for human GWAS prioritization.

28. Please remove the CICERO-inferred co-accessibility of open chromatin regions from the ATAC-seq pictures. The interaction of regulatory elements needs to be shown experimentally with 3D chromatin conformation assays such as Hi-C, 3C, 4C

Thank you. While we agree that CICERO is a computational tool to understand co-accessibility, we believe that it provides valuable information on co-accessibility. Indeed, at this moment there is no perfect method to prove co-accessibility. Hi-C has several shortcomings, including its resolution. To this end, we kept only some of the CICERO-based analysis but commented on its limitations.

29. The last sentence includes reference to disease development. Data are derived from uninjured kidney tissue. The links to disease states are not developed in the manuscript and any reference to disease states should be removed from the manuscript.

Thanks for pointing this out. We were referring to human disease related genes in GWAS. We have rephrased the sentence.

Reviewer #4 (Remarks to the Author):

This this is an outstanding paper that assesses the chromatin accessibility at a single cell/single nucleus level for developing mouse kidneys and adult kidneys. Coupled with scRNA-seq, the paper characterizes cell-type-specific DNA regulatory elements that could provide insights into kidney development and differentiation. The data allows identification of renal epithelial cell type specific enhancers and predictions of the transcription factors bound in these enhancers. The authors provide their curated data in the form of three publicly accessible web pages that will garner heavy use. These data will be extremely valuable to the research community. The following are intended to be constructive comments useful in revising the paper.

We thank the reviewer for the positive comments on our manuscript. We hope our manuscript could be useful for the whole community.

Semi-Major

1. In the cell type annotation stage, proximal tubule cells were classified into S1 and S3 cells in both scATAC-seq and scRNA-seq. Where are the S2 cells? Kap transcripts seem to be abundant in both S2 and S3 cells [PMID: 25817355 and 31689386]. In this regard, the authors' S3 could be a combination of S2 and S3. S3 cells are rare compared to S2 because they are only found in the part of the proximal tubule in the outer stripe of the outer medulla. The classic physiological function of S2 is PAH secretion [PMID: 659594]. In this regard, the PAH transporter *Slc22a6* could be of value.

Thank you for this important comment. Identification of the S2 PT segments has proved challenging in scRNA-seq studies with few specific S2 marker genes available in the single cell RNA-seq datasets (Park et al., Science 2018).

We performed subclustering analyses of PT cells in the scRNA-seq data. We were not able to delineate distinct S2 identity markers, even when checking for markers known to be associated more with S2 than with S3 (e.g. *Slc22a12*, *Slc22a7*, *Slc27a2*, *Slc7a13*) (**Additional Figure 12**). In this respect, it was interesting to see that *Slc22a6* was most highly expressed in the cells that strongly express typical S3 markers (such as *Slc22a30*, *Cyp4b1*, *Atp11a*). Even by over-clustering the data, a clear S2 identity could not be defined. It is likely that S2 and S3 cells are very similar to each other in gene expression space. We therefore relabeled the identity of these clusters as proximal tubule convoluted segment (PCT) and proximal tubule straight segment (PST) throughout the manuscript, figures, supplemental figures and tables, respectively.

Additional Figure 12. Violinplots of PT markers known to be associated with the S2 segment.

It is interesting to note that PT subclustered into PCT and PST in the snATAC-seq dataset rather than S1, S2 and S3 segments, indicating clear differences between PCT and PST segments globally, but less differences between S2 and S1 and S3 segments. We also relabeled these cell types as PCT and PST.

2. Different cell type names were interchangeably used throughout the main figures and supplementary figures. For example, PTS1 and PTS2 v.s. PT, PT_out, and PT2. This is confusing. Consistent naming of cells/cell types is desirable.

Based on the analysis above, we relabeled the cluster to proximal tubule straight segment (PCT) and proximal tubule convoluted segment (PST) throughout the manuscript, figures, supplemental figures and tables, respectively.

The PT_out cluster in the snATAC-seq is likely an intermediate cell type between NP and PT (**Figure S16b**), so we renamed this cluster as “early PT”. However, in our downstream analysis, we only focused on the well-studied PCT and PST groups.

3. It seems that the authors ignored the ureteric bud-derived cells (CNT, PC, IC) in the analysis associated with Fig. 3. Are there specific reasons for this? As the author mentioned in the introduction, these cells differ from metanephric mesenchyme derived cells. how were these cells ordered in Fig. 3? Did they show a different trajectory?

Thank you for this comment. Our dataset did not capture the ureteric bud cells with high fidelity, we were therefore not able to perform trajectory analyses for UB, we felt that including

ureteric bud-derived cell types into the NP-derived trajectory would complicate the analysis, for this reason we did not include CNT, PC and IC in Fig. 3.

4. Line 361, it is unclear how NP1-NP3 were defined. What is the basis for separating the NP into Np1-NP3? Without a clear definition in the text, it leads to confusion. The same issues were found related to Figure 4. This is an interesting figure, however, multiple new terminologies and cell types are mentioned without clear explanation and justification. Thank you for this comment. Per the suggestion, we clarified this in the text and conducted additional analysis to investigate the identity of these groups (**Figure 4b**).

The cell differentiation data was generated using the unbiased cell trajectory analysis, such as Monocle and RNA velocity (**Figures 3 and S16**). This analysis highlighted important bifurcation steps and stages that show differences based on the open chromatin data. We then performed differential accessibility analysis and gene ontology analysis for these stages and this information is presented in **Figure 4**. Our results indicate some new stages of kidney development such as NP1-3 etc., and some that are similar to previous analysis. These terminologies reflected the order of cellular differentiation based on our computational analysis.

To investigate the identities for the different groups, we studied the gene activity score enrichment of prior annotations including renal vesicle and S-shaped body (**Figure 6b**). Specifically, NP3 is matched to renal vesicles based on the enrichment of *Wnt4* (Park et al., Dev 2007), IM1-2 and Podo1 match Comma- and S- shaped body signatures, such as *Notch1*, *Notch2*, *Jag1*, and *Lfng* (Lindstrom et al., J Am Soc Nephrol 2018). Thus IM1-2 and Podo1 may represent CSB/SSB that committed to tubule or podocyte lineages, respectively. However, these nephron progenitor subgroups (or states) need to be functionally validated.

Recently, a new paper (Lindstrom et al., Biorxiv 2020) aims to map nephron differentiation using spatial marker mapping, which could provide additional insight into kidney development.

An important point we aimed to emphasize was that the open chromatin data was able to identify developmental stages in much greater fidelity than transcriptomics data, while this might be technology related but likely also speaks for the importance of the epigenome in defining cell types.

5. Line 522, the statement “the GWAS locus demonstrated a strong open chromatin region in nephron progenitors but not in any other differentiated cell types”, accessible chromatin regions are seen at *Uncx* locus in Loop of Henle (Fig. 1D and Fig. 6E), but the weak signals could be simply “averaged” due to the fact that multiple cell types with distinct functions are present in Loop of Henle region (thin descending limb, thin ascending limb, medullary thick ascending limb, etc.). Did the author try to delineate the cell types in LOH region?

Thanks for pointing this out. We found that the signal in the LOH comes from the two regions: chr5:139543458-139544188 and chr5:139547063-139547458. Investigating the accessibility in UMAP, we found that only sporadic cells in LOH show accessible in this region, and the

magnitude was similar to other cells including podocytes. By contrast, the signal was enriched in NP (**Additional Figure 13**). We reason that the sporadic accessibility could be due to misclassification or read contamination.

Consistently, we only observed *Uncx* expression in nephron progenitors in the scRNA-seq data. Furthermore, we found publications have verified that *Uncx* was expressed in the nephron progenitors in mouse and human (Neidhardt et al, *Dev. Genes Evol.* 1997; Hochane et al, *PLOS Biol.* 2019). To this end, we believe *Uncx* gene expressed and accessibility is enriched in the nephron progenitors.

Additional Figure 13. Accessibility feature plot of two peaks around LOH region

6. In viewing the scRNA-seq data at susztaklab.com/developing_adult_kidney/scRNA/, it appears that *Aqp1*, *Aqp2*, and *Aqp3* have signals in most clusters suggesting that there is a lot of ambient mRNA in the samples. I think, however, that this is just a scaling problem. Would advise setting some threshold value below which cells are reported with gray color rather than dark blue.

We thank the reviewer for this important comment. Indeed, ambient RNA is a problem in scRNA-seq datasets, which is under-recognized in scRNA-seq data analysis so far.

We therefore re-analyzed the RNA-seq dataset after ambient RNA-induced signal was cleaned using the R package SoupX and found that correction for ambient RNA did not change downstream results. Quantification of the effect of ambient RNA on every single cell in the dataset can be viewed for P0 and adult tissues separately in **Figures S6a-p**. For the examples mentioned by the reviewer (*Aqp1*, *Aqp2*, *Aqp3*) we show that gene expression is similar after ambient RNA correction (**Figures S7a-b**). Although SoupX estimated a considerable fraction of cells to be contaminated by ambient RNA (10.3% in Adult, 15.8% in P0), re-running our whole analysis pipeline with an ambient RNA-corrected count matrix yielded a very high correlation of cluster gene averages when comparing ambient RNA-cleaned and -uncleaned matrices (**Figure S7c**).

We provide the highly identical list of DEGs for the respective clusters calculated from the ambient RNA-corrected count matrix as Suppl. Table 12. We have included respective paragraphs to the methods and results sections.

In addition, we agree with the reviewer that the visualization on our website http://susztaklab.com/developing_adult_kidney/scRNA/ might be non-intuitive using a viridis scale. Users could adjust the color scale in the upper right panel, as indicated in **Additional Figure 14**.

Additional Figure 14. Settings in the visualization tool to change color scale

7. While the data at susztaklab.com/developing_adult_kidney/snATAC/ are potentially very useful, searching on the basis of the bin coordinates seems impractical. It would be very nice if a bin could be selected by specifying the official gene symbol of one of the genes whose TSS is found in the bin.

Thank you. We have adjusted the website and now researchers can also search the gene symbols to get the chromatin accessible landscape. Searching by genomic location is still available as it is also important.

We have also made the data publicly available in GEO with accession number GSE157079, so expert users can download and reanalyze the data.

.... Minor

1. Line 155, “Sc12a3” should be “Slc12a3”.

Thank you, we corrected it.

2. The definition of “Stroma cell” is broad. It could be resident fibroblast, vascular smooth muscle, mesangial cells, etc. Since the authors have generated the differential genes/peaks, this could be elaborated. In Fig. 1F, Were stroma 1 and 2 combined?

We thank the reviewer for this question. We refer this reviewer to the response to the comment of reviewer#3, point 4, where we highlight our re-analysis of the stroma clusters, demonstrating their heterogeneity (new **Figures S8 and S9**). For simplicity reasons, here we combined Stroma 1 and 2 in Figure 1f.

3. In Fig. 2a, PU.1 and Spi1 were interchangeably used.

Thank you. PU.1 is the name of transcription factor, which is identified by Homer analysis, while *Spi1* is the gene symbol for this transcription factor.

4. Fig. S1-I, the cell annotation and values are not aligned correctly. For example, It is not clear which value was assigned to PC, etc.

Thank you for noticing, we fixed this. Also, **Figure S5a** (former **Figure S1i**) now reports cell numbers and percentages grouped by Adult and P0.

5. Will the author provide the bulk adult ATAC-seq track in

http://susztaklab.com/developing_adult_kidney/igv/?

Thank you, the bulk tracks are now included in the IGV website.

6. Line 368, which “Notch” are the authors referring to?

Thank you for noticing this detail. *Notch2* decreased between stages Podo1 and Podo2, whereas *Notch1* decreased between Podo2 and Podo3. We have adjusted the text to include the specific information.

7. Line 388, the statement “we performed immunofluorescence studies on developing kidneys (E13.5, P0, and P6)”, the author provided IF for E13.5 and P6, but P0 was missing in main and supplementary figures.

Thank you for noticing. We now include IF staining of P0 in Figures S19 and S20. Figure S19 demonstrates FOXL1 staining in S-shaped bodies and glomeruli. Figure S20 highlights FOXL1 co-localization with WT1 and JAG1 positive podocytes in glomeruli.

8. The ligand-receptor analysis is interesting, but could be moved to supplementary figures as the main focus of the paper is chromatin accessibility.

Thank you, we simplified this section.

9. recommend: 'lowly expressed' -- 'expressed at low levels'

Thank you, we made this change throughout the manuscript.

10. The Introduction states, "The distal convoluted tubule is critical for regulated electrogenic sodium reabsorption and potassium secretion." This statement is indirectly true but likely to be misunderstood. The major route of Na reabsorption is electrically neutral since it is mediated by the Na-Cl cotransporter (NCC, Slc12a3) of the DCT and the apical plasma membrane lacks parallel ion channels. Since no potassium pathway has been identified in the apical plasma membrane of the DCT, the DCT is not believed to mediate net transepithelial K transport. It

indirectly affects electrogenic transport in the collecting duct by altering delivery of Na to the principal cells.

Thank you, we made this change.

Mark Knepper

REVIEWER COMMENTS

Reviewer #1 (Remarks to the Author):

The authors have been generally responsive to my comments in their rebuttal. However, I would have liked to see a little bit more of these comments translating into changes in the manuscript. For example, the response to comment 3 and 5 should be more explicit in Discussion.

Reviewer #2 (Remarks to the Author):

I have reviewed the response to reviewers and the updated manuscript by Miao and colleagues. Overall, I am largely satisfied with the response and there are mainly minor things that I would recommend changing.

The main section that I still think overstates the case is the GWAS section. For example, the title of this section (lines 437-438) currently reads: "Single cell chromatin accessibility identifies human kidney GWAS target regulatory regions, genes and cell types". The data simply do not support this. The implications of this section are enticing. And the potential of following up these leads is exciting. But there are too many caveats to the analysis to frame it this way without having done those significant follow up studies. The section title should read something like "Murine single cell chromatin accessibility implicates potential human kidney GWAS mechanisms, target regulatory regions, genes and cell types". The first sentence of this section (line 440-441) should replace "identify" with "implicate", or something similar. The sentence in lines 446-448 should soften as well, with something like: "Here, we reasoned that murine single cell accessible chromatin information could be useful in nominating potential cell type-specific regulatory regions and thereby the target cell type for the GWAS hits" (adding "murine", and "in nominating potential"). Given that only a small fraction of SNPs lift over and we don't know how well the regulatory landscape is conserved, I don't find the sentence in lines 481-482 ("Importantly, only a small subset of GWAS significant SNPs overlapped with open chromatin areas.") persuasive. I would remove it. I would also encourage the authors to be a little more specific about the mechanism they think is plausible (using cautious language) in the Shroom3 case. Do they think that disrupting Six2 binding in a nephron progenitor may lead to improper development and ultimate kidney function disorder? On lines 488-489, without a principled statistical analysis that takes into account confounders (such as the number of peaks identified in each cell type, the location of those peaks relative to genes, the GC content of those peaks, the mappability of those peaks, etc.), I would remove this sentence. In addition, I didn't follow how the figure cited here (S23) supported a single cell type for each SNP. I see an apparent enrichment in cell types, but it doesn't look exclusive to me. In line 490-493, I would add the term "murine" – i.e. "variants associated with kidney disease development are located in regions with murine cell type- and developmental stage-specific regulatory activity" – and change "critical role" to "potential power". In the discussion, the language also needs to be modified around this topic. The sentence on line 538-539 should read something like "Furthermore, we show that murine single cell- and differentiation stage-level epigenome annotation is potentially powerful for the annotation of human GWAS." (adding "murine" and "potentially powerful"). For the sentence on lines 541-543, the word "identification" should be replaced by something more careful such as "implication". The next sentence (lines 543-546), the word "those" should be replaced by "murine". The next sentence (lines 546-548) should say "important potential mechanism" instead of "important mechanism". The next sentence (lines 548-550) should say the "This mechanism would be similar" not "is similar".

With respect to a second point I made about citing and comparing previous relevant literature, I would encourage the authors to at least mention the comparison with previous single cell data in the manuscript. The previous work used different methods for cell/nuclei isolation and different single-cell technologies and so it would be helpful for the field to know how they stack up in terms of QCing the data. Especially as this study presents an improvement on the ability to identify known cell types (even if this is in part because of profiling more cells).

Finally, there are few minor issues that the revisions introduced:

Line 128 – the authors refer to “UMIs” for scATAC-seq data. Technically, the “UMI” of 10X scATAC data is the insertion site. I think some readers may find this confusing. I would recommend just saying “filtering of the number of unique fragments” or something along those lines.

Line 134-135 – The authors refer to clusters as cell types (“nephron progenitors and stromal cells”) before mentioning how clusters were annotated as specific cell types. I would move this sentence to after the cell type annotation description.

Line 164 – Similarly, Fig. S4a is referenced (which shows data stratified by cell types) before the scRNA-seq cell types are annotated in the text. I would move this reference until later and possibly re-order the panels in S4 if that makes sense.

Lines 166-168 – Fig. S5 is cited before S4f-g.

Lines 278-282 – Fig. S15 is referenced before S14b-c.

Line 290 – I couldn't follow how Fig. S11d would support the statement. Do the authors mean S11b?

Lines 458-461 – Fig. 6c is cited before Fig. 6b.

Lines 466-470 – Fig. 6f is cited before Fig. 6e.

Lines 481-485 – Figs. 6h and 6i are not cited in the text. I would incorporate them somewhere in here.

Line 485 – the panels in Fig. S22 are in a different order than 6. I would recommend consistency. Also, some of the peaks that are apparent in S22 can't be seen in 6. It might be worth replacing 6 with S22.

Line 988,992 – I believe the function in Monocle is “differentialGeneTest”, but it's coming out as two words in the Methods. This could lead to confusion. Please correct.

Reviewer #3 (Remarks to the Author):

The authors have responded in an excellent way to the reviewers' comments. This is an outstanding manuscript

Joseph Bonventre

Reviewer #4 (Remarks to the Author):

Outstanding work. Thanks for your detailed responses.

Reviewer #1 (Remarks to the Author):

The authors have been generally responsive to my comments in their rebuttal. However, I would have liked to see a little bit more of these comments translating into changes in the manuscript. For example, the response to comment 3 and 5 should be more explicit in Discussion.

We thank the reviewer for the suggestions. We included a respective sentence on the need for human tissues in the Discussion section.

For your comment 5, we acknowledge that snATAC-seq data could not faithfully overcome the issue of LD structure, so we have deleted the section pertaining to LD structures.

Reviewer #2 (Remarks to the Author):

I have reviewed the response to reviewers and the updated manuscript by Miao and colleagues. Overall, I am largely satisfied with the response and there are mainly minor things that I would recommend changing.

We thank the reviewer for the recognition of our responses. We adjusted our manuscript per your suggestions.

The main section that I still think overstates the case is the GWAS section. For example, the title of this section (lines 437-438) currently reads: "Single cell chromatin accessibility identifies human kidney GWAS target regulatory regions, genes and cell types". The data simply do not support this. The implications of this section are enticing. And the potential of following up these leads is exciting. But there are too many caveats to the analysis to frame it this way without having done those significant follow up studies. The section title should read something like "Murine single cell chromatin accessibility implicates potential human kidney GWAS mechanisms, target regulatory regions, genes and cell types".

We changed the section title accordingly.

The first sentence of this section (line 440-441) should replace "identify" with "implicate", or something similar.

We made this edit.

The sentence in lines 446-448 should soften as well, with something like: "Here, we reasoned that murine single cell accessible chromatin information could be useful in nominating potential cell type-specific regulatory regions and thereby the target cell type for the GWAS hits" (adding "murine", and "in nominating potential").

We rephrased the sentence accordingly.

Given that only a small fraction of SNPs lift over and we don't know how well the regulatory landscape is conserved, I don't find the sentence in lines 481-482 ("Importantly, only a small

subset of GWAS significant SNPs overlapped with open chromatin areas.”) persuasive. I would remove it.

Thank you, we deleted the sentence.

I would also encourage the authors to be a little more specific about the mechanism they think is plausible (using cautious language) in the Shroom3 case. Do they think that disrupting Six2 binding in a nephron progenitor may lead to improper development and ultimate kidney function disorder?

Thank you for this comment. We think the reviewer agrees with our notion that any further comment on the mechanism in the Shroom3 case is purely speculative. While the fact that one SNP with strong nephron progenitor-specific enrichment coincided with the Six2 binding area is enticing, uncovering mechanistic insights definitely warrants experimental validation. We have added a corresponding statement to this part of the discussion to highlight this fact.

On lines 488-489, without a principled statistical analysis that takes into account confounders (such as the number of peaks identified in each cell type, the location of those peaks relative to genes, the GC content of those peaks, the mappability of those peaks, etc.), I would remove this sentence.

We deleted the sentence.

In addition, I didn't follow how the figure cited here (S23) supported a single cell type for each SNP. I see an apparent enrichment in cell types, but it doesn't look exclusive to me.

Thank you. As suggested, this analysis is simply an overlap analysis, so the results should be interpreted with caution. We used 10% as a cutoff to determine if the peak is “accessible” in one cell type, however, we deleted this figure (Figure S23) in this revision.

In line 490-493, I would add the term “murine” – i.e. “variants associated with kidney disease development are located in regions with murine cell type- and developmental stage-specific regulatory activity” – and change “critical role” to “potential power”.

We made these changes.

In the discussion, the language also needs to be modified around this topic. The sentence on line 538-539 should read something like “Furthermore, we show that murine single cell- and differentiation stage-level epigenome annotation is potentially powerful for the annotation of human GWAS.” (adding “murine” and “potentially powerful”). For the sentence on lines 541-543, the word “identification” should be replaced by something more careful such as “implication”. The next sentence (lines 543-546), the word “those” should be replaced by “murine”. The next sentence (lines 546-548) should say “important potential mechanism” instead of “important mechanism”. The next sentence (lines 548-550) should say the “This mechanism would be similar” not “is similar”.

We adjusted the sentences accordingly.

With respect to a second point I made about citing and comparing previous relevant literature, I would encourage the authors to at least mention the comparison with previous single cell data

in the manuscript. The previous work used different methods for cell/nuclei isolation and different single-cell technologies and so it would be helpful for the field to know how they stack up in terms of QCing the data. Especially as this study presents an improvement on the ability to identify known cell types (even if this is in part because of profiling more cells).

We thank the reviewer for this suggestion. We agree that such comparison will be beneficial for the field. We also agree the higher resolution of our dataset could stem from the fact that we have more cells profiled, or technology difference. To this end, we presented the comparison of our dataset with the Mouse ATAC Atlas dataset in the **Methods** section (line 805). However, we would like to point out that there is no dataset of the developing kidney encompassing both scRNA-seq and snATAC-seq data.

Finally, there are few minor issues that the revisions introduced:

Line 128 – the authors refer to “UMIs” for scATAC-seq data. Technically, the “UMI” of 10X scATAC data is the insertion site. I think some readers may find this confusing. I would recommend just saying “filtering of the number of unique fragments” or something along those lines.

We replaced the ambiguous word.

Line 134-135 – The authors refer to clusters as cell types (“nephron progenitors and stromal cells”) before mentioning how clusters were annotated as specific cell types. I would move this sentence to after the cell type annotation description.

We moved the sentence to the end of next paragraph.

Line 164 – Similarly, Fig. S4a is referenced (which shows data stratified by cell types) before the scRNA-seq cell types are annotated in the text. I would move this reference until later and possibly re-order the panels in S4 if that makes sense.

Thank you for noticing, we made edits in the main text so that Fig. S4a is mentioned after kidney cell types are introduced.

Lines 166-168 – Fig. S5 is cited before S4f-g.

We reorganized the sentence.

Lines 278-282 – Fig. S15 is referenced before S14b-c.

We reorganized the figures.

Line 290 – I couldn't follow how Fig. S11d would support the statement. Do the authors mean S11b?

Thank you. Fig. S11d shows the number of peaks that are unique to NP is more than the number of peaks that are unique in Podo or LOH. Although the number of unique peaks in PT is higher than that in NP, when we sum up the number of peaks shared between NP and Podo (the fifth column) and peaks shared between NP and LOH (the seventh column), the total number of accessible peaks in NP is still higher than that in PT. In addition, the PT has a much larger population compared with other cell types, which might be the reason of relatively more peaks. In general, this comparison shows that the chromatin closing is more prevalent than

chromatin opening during differentiation. Interestingly, we noticed that another bioRxiv preprint showed similar findings from blood cells that pluripotent cells have more open chromatin (Ranzoni et al., bioRxiv 2020 doi: <https://doi.org/10.1101/2020.05.06.080259>).

Lines 458-461 – Fig. 6c is cited before Fig. 6b.

Lines 466-470 – Fig. 6f is cited before Fig. 6e.

Thanks for pointing this out. We relabeled the figures. We kept mouse tracks together so people could match the open chromatin peaks with the epigenome tracks.

Lines 481-485 – Figs. 6h and 6i are not cited in the text. I would incorporate them somewhere in here.

Thank you for noticing, we added the citations.

Line 485 – the panels in Fig. S22 are in a different order than 6. I would recommend consistency.

Thank you, we adjusted the order of Fig. S22 so that both Figures are consistent.

Also, some of the peaks that are apparent in S22 can't be seen in 6. It might be worth replacing 6 with S22.

Thank you. Figure S22 show some examples of SNPs that are consistent in both mouse and human, but it is unknown whether these SNPs are more likely to be causal than other SNPs. To this end, we keep Figure S22 in the supplemental figures.

Line 988,992 – I believe the function in Monocle is “differentialGeneTest”, but it's coming out as two words in the Methods. This could lead to confusion. Please correct.

Thank you for noticing, we fixed this.

Reviewer #3 (Remarks to the Author):

The authors have responded in an excellent way to the reviewers' comments. This is an outstanding manuscript

We thank the reviewer for the positive comments of the manuscript and our response.

Joseph Bonventre

Reviewer #4 (Remarks to the Author):

Outstanding work. Thanks for your detailed responses.

We thank the reviewer for the positive comments of the manuscript and our response.

REVIEWERS' COMMENTS

Reviewer #2 (Remarks to the Author):

At this point, I'm satisfied with the responses of the authors. I appreciate the work they did to address my concerns.

Reviewer #2 (Remarks to the Author):

At this point, I'm satisfied with the responses of the authors. I appreciate the work they did to address my concerns.

We thank the reviewer for the comments.